# GLOBAL RESOLUTION: OPTIMAL MULTI-DRAFT SPECULATIVE SAMPLING VIA CONVEX MINIMIZATION

**Rahul Krishna Thomas**[1,2*]
[1]Ritual      [2]Columbia University

**Arka Pal**[1]
[1]Ritual

## ABSTRACT

Speculative sampling reduces the latency of autoregressive decoding for target model LLMs without sacrificing inference quality, by using a cheap draft model to suggest a candidate token and a verification criterion to accept or resample this token. To improve acceptance and decoding efficiency, recent work has explored the multi-draft extension, where at each step $n$ draft tokens are generated, and the verification criterion is a distribution conditioned on these. When this criterion maximizes the probability of accepting some draft token, it is called the optimal transport (OT). However, finding the OT is difficult, as it is the solution of a linear program (OTLP) in over $V^n$ variables, with $V$ being the vocabulary size. Two recent theoretical works have reframed the OTLP in terms of importance sampling or subset selection. In this work, we prove that these formulations are equivalent to an exponentially large relaxed OTLP, so it remains infeasible to solve. Then, we reverse engineer subset selection to formulate the OTLP as a max-flow problem. With a novel application of polymatroid theory, we reduce the exponentially large OTLP to a convex optimization problem in at most $V$ variables. This allows us to devise an algorithm for optimal $n$-draft speculative sampling when the $n$ tokens are chosen i.i.d. from a single draft model, which can be tuned to arbitrary accuracy. Finally, we measure acceptance rates and algorithm runtimes for various $n$ and top-$k$ draft sampling settings. Our findings give the first multi-draft algorithm with 90% acceptance and under 100 ms of overhead per generated token with negligible deviation from the target model distribution.

## 1 INTRODUCTION

Language models have demonstrated state-of-the-art performance over a wide variety of tasks, such as code generation, language processing, and complex reasoning (Zhu et al., 2024; Kasneci et al., 2023; Thirunavukarasu et al., 2023). Many transformer-based language model families, like GPT (Radford et al., 2018; 2019; Brown et al., 2020; OpenAI, 2023) and Llama (Touvron et al., 2023b;a), use autoregressive decoding, where tokens are generated sequentially. However, autoregressive decoding leads to significant computational bottlenecks. For LLMs with hundreds of billions of parameters, these bottlenecks are generally dominated by memory bandwidth, not raw FLOPs (Fu et al., 2024).

Speculative sampling was proposed by Chen et al. (2023); Leviathan et al. (2023) to reduce the punitive costs of autoregressive decoding for an expensive target model. Motivated by speculative execution in compilers, this uses a light draft model to generate a draft sequence. Each token of the draft sequence is then accepted or resampled using a criterion, which ensures that the first resampled token and all prior accepted tokens follow the true target distribution. The greatest speedup is obtained by maximizing the acceptance probability, as accepted tokens are generated with essentially no inference overhead.

Recent extensions of speculative sampling have fixed the draft sequence length to one (**single-step**) and focused on the theoretical properties of the **multi-draft** setting. In this setting, $n \geq 2$ draft tokens are generated at the single draft sequence position, for example by sampling from one or more draft models. The $n$ corresponding target logits are efficiently computed in parallel, adding minimal overhead to a single target forward pass for reasonable batch sizes $n$. Finally, a conditional

---

*Correspondence to: `rahulkrishnathomas@gmail.com`.

distribution over the $n$ draft tokens (verification criterion) generates a verified token, which must respect the true target distribution. This conditional distribution is called the **optimal transport (OT)** when the verified token lies among the $n$ draft tokens with maximal probability, thereby giving the optimal acceptance probability and speedup.

The first theoretical work in the multi-draft setting was by Sun et al. (2023), who proved that the OT is a solution to an **optimal transport linear program (OTLP)**. This LP has a number of variables exponential in the vocabulary size, so it is not even feasible to write down for large vocabularies. However, Khisti et al. (2025) recently compactified this into a two-step canonical decomposition: importance sampling and *single-draft* speculative sampling. When $n = 2$ draft tokens are sampled from one draft model, this leads to an explicit expression for the optimal acceptance probability. Concurrently, Hu et al. (2025) has reframed optimal acceptance as the solution to a subset selection problem, which matches this $n = 2$ expression and generalizes it to $n \geq 3$ draft tokens.

While these works can compute optimal acceptance for $n \geq 2$, they do not solve the OTLP. There is therefore, to the best of our knowledge, no exact *and* efficient method to solve the OTLP and find the optimal verification criterion in multi-draft speculative sampling. The only known exact method is an LP solver, which has exponential time complexity for the OTLP. Existing *approximation* algorithms such as K-SEQ (Sun et al., 2023), MSS (Miao et al., 2024), RSD (Jeon et al., 2024), and LP and vocabulary truncation (Khisti et al., 2025) cannot guarantee near-optimal speedups: K-SEQ has a $(1 - 1/e)$-approximation guarantee, but the others have no formal guarantees.

**Our main contribution is the first efficient algorithm to compute the OT to arbitrary accuracy when $n \geq 2$ draft tokens are i.i.d. sampled from a draft model, in the single-step multi-draft setting.** Our algorithm is fast for much larger vocabularies and choices of $n$ than previous works, permitting higher acceptances rates. Altogether, our theoretical and experimental contributions are:

- We show how the canonical decomposition (Khisti et al., 2025) of the OTLP is exactly equivalent to LP relaxation in the derivation of the subset selection formulation (Hu et al., 2025), unifying the current state-of-the-art in multi-draft theoretical literature.

- We reverse the derivation in (Hu et al., 2025) to obtain a max-flow algorithm to compute the OT for arbitrary $p_{\text{draft}}$. While the max-flow network is enormous, one can compute feasible flows along source edges in near-linear time with a greedy polymatroid algorithm. With these flows set, the OT can be computed to arbitrary accuracy by solving a truncated convex minimization problem. This leads to our **global resolution** OT algorithm in the i.i.d. setting.

- We compute optimal acceptance rates and algorithm runtimes for various choices of $n$ and top-$k$ sampling of the draft model in the i.i.d. setting. Our algorithm permits practical choices of $k$ and $n$ where acceptance rates are over 90% and average runtime is under 100 ms/token, whereas generic OTLP solvers can only achieve 84% in the same limit. Thus, we obtain a new state-of-the-art in *optimal* multi-draft speculative sampling.

## 2 RELATED WORK

To improve the number of accepted tokens, most theoretical extensions of speculative sampling have been split into *multi-step* and *multi-draft*, which alter the verification or drafting mechanisms.

In the multi-step setting, which was outlined in the first few speculative sampling works (Chen et al., 2023; Leviathan et al., 2023), the draft model autoregressively generates a *sequence* of candidate tokens, and accepts the longest *prefix* of tokens that pass verification. Some work has fixed verification as independent single-step verification, and focused on improving acceptance through smarter drafting, such as database retrieval (He et al., 2023), cascading (Chen et al., 2024), hierarchial drafting (Sun et al., 2024a), knowledge distillation (Zhou et al., 2023b; Liu et al., 2023), layer skipping (Zhang et al., 2023; Elhoushi et al., 2024), or multi-token sampler heads Gloeckle et al. (2024); Samragh et al. (2025). Other work has developed new verification protocols, such as tree Monte Carlo (Hu & Huang, 2024) and block verification (Sun et al., 2024b), with the latter shown to be naturally optimal.

While most multi-step works are based on single-draft speculative sampling, tree-based attention frameworks (Sun et al., 2023; Spector & Re, 2023; Cai et al., 2024; Li et al., 2024) extend to the multi-draft setting by creating a draft *tree* of token sequences. Our work instead focuses on the **single-step, multi-draft** setting, which is key to improving these frameworks. For example, SpecTr

(Sun et al., 2023) can use *any* single-step, multi-draft method, e.g. an approximate OTLP solver like K-SEQ. By improving the single-step OTLP solver, we see efficiency gains in multi-step SpecTr.

## 3 BACKGROUND ON MULTI-DRAFT SPECULATIVE SAMPLING

For a review of multi-draft speculative sampling, see Appendix A. Here, we review the theory of single-step multi-draft speculative sampling, which is primarily based on the OTLP (Sun et al., 2023) and its subset selection formulation (Hu et al., 2025).

### 3.1 THE OPTIMAL TRANSPORT LP

Many multi-draft procedures respect the target model distribution (e.g. K-SEQ), but we are interested in the one with greatest speedup. In the single-step setting, the parameters of this optimal sampling procedure are solutions to an **optimal transport linear program (OTLP)** (Sun et al., 2023). In terms of the number of draft models $n$, the target model distribution $p$ over the vocabulary $\mathcal{V}$, and the $n$-token draft distribution $p_{\text{draft}}$ over $\mathcal{V}^n$, the OTLP has $V^{n+1}$ nonnegative variables $C_{i,\bar{i}}$ for $i \in \mathcal{V}, \bar{i} \in \mathcal{V}^n$ and is written as:

$$\max_{\boldsymbol{C} \succeq 0} \sum_{i \in \mathcal{V}} \sum_{\bar{i} \in A_i} C_{i,\bar{i}} \quad \text{s.t.} \quad \sum_{\bar{i} \in \mathcal{V}^n} C_{i,\bar{i}} = p(i) \quad \forall i \in \mathcal{V}, \quad \sum_{i \in \mathcal{V}} C_{i,\bar{i}} = p_{\text{draft}}(\bar{i}) \quad \forall \bar{i} \in \mathcal{V}^n. \quad (1)$$

Here, $A_i = \{\bar{i} \in \mathcal{V}^n : i \in \text{set}(\bar{i})\}$ is the incidence set of each $i \in \mathcal{V}$ over all drafted $n$-tuples. The optimal objective value is the **optimal acceptance rate** $\alpha^*$, and the optimal parameters $C^*$ give a joint distribution $\pi(i, \bar{i})$ over $\mathcal{V} \times \mathcal{V}^n$ with marginals $p, p_{\text{draft}}$. The **optimal transport (OT)** is the induced conditional distribution $\pi(i|\bar{i})$, which gives the probability of verifying $i \in \mathcal{V}$ given $n$ draft-sampled tokens $\bar{i} \in \mathcal{V}^n$. Solving the OTLP in this form is infeasible due to its exponential size, even for $n = 2$ and $V$ as small as 1000: see Section 7.

### 3.2 SUBSET SELECTION FORMULATION

Computing the optimal objective of the OTLP can be far simpler than solving it. Hu et al. (2025) show this by reducing the optimal acceptance computation to subset selection (Equation (4)) for any $p_{\text{draft}}$. This allows them to bypass the OTLP and quickly solve for $\alpha^*$ for some choices of $p_{\text{draft}}$ (see Section 5.1). To do this, they form a **relaxed OTLP** in nonnegative variables $S_{i,\bar{i}}$ for $i \in \mathcal{V}, \bar{i} \in \mathcal{V}^n$:

$$\max_{\boldsymbol{S} \succeq 0} \sum_{i \in \mathcal{V}} \sum_{\bar{i} \in \mathcal{V}^n} S_{i,\bar{i}} \quad \text{s.t.} \quad \sum_{\bar{i} \in \mathcal{V}^n} S_{i,\bar{i}} \leq p(i) \quad \forall i \in \mathcal{V}, \quad \sum_{i \in \mathcal{V}} S_{i,\bar{i}} \leq p_{\text{draft}}(\bar{i}) \quad \forall \bar{i} \in \mathcal{V}^n,$$
$$S_{i,\bar{i}} = 0 \quad \forall i \in \mathcal{V}, \bar{i} \notin A_i. \quad (2)$$

One can get an OTLP solution $\boldsymbol{C}^*$ from a relaxed OTLP solution $\boldsymbol{S}^*$ with *residuals*:

$$C_{i,\bar{i}}^* = S_{i,\bar{i}}^* + \frac{p^{\text{res}}(i) p_{\text{draft}}^{\text{res}}(\bar{i})}{\sum_{j \in \mathcal{V}} p^{\text{res}}(j)}, \quad p^{\text{res}}(i) = p(i) - \sum_{\bar{i} \in \mathcal{V}^n} S_{i,\bar{i}}^*, \quad p_{\text{draft}}^{\text{res}}(\bar{i}) = p_{\text{draft}}(\bar{i}) - \sum_{i \in \mathcal{V}} S_{i,\bar{i}}^*. \quad (3)$$

They then dualize the relaxed OTLP and simplify it to the following. We defer details to Appendix E.

**Theorem 3.1** (Hu et al. (2025)). *The optimal acceptance rate $\alpha^*$ can be computed as*

$$\alpha^* = 1 + \min_{H \subseteq \mathcal{V}} \psi(H), \quad \psi(H) = \sum_{i \in H} p(i) - \sum_{\bar{i} \in H^n} p_{draft}(\bar{i}). \quad (4)$$

## 4 CANONICAL DECOMPOSITION BY RELAXED LP

The main result of Khisti et al. (2025) is the **canonical decomposition** of the OT: it can be split into importance sampling and single-draft speculative sampling. First, after sampling the $n$ draft tokens $\bar{i} \sim p_{\text{draft}}$, one samples a token in set($\bar{i}$) using an importance-weighted distribution $\beta(i|\bar{i})$. Then, one applies single-draft speculative sampling from Chen et al. (2023); Leviathan et al. (2023) on $i$. Canonical decomposition thus computes the OT by solving a non-linear $\beta$-optimization problem.

In this section, we prove that solving the $\beta$-optimization problem is mathematically equivalent to solving the relaxed OTLP from Equation (2). The optimal importance sampling parameters in the canonical decomposition can be formed by normalizing the optimal relaxed LP parameters $\boldsymbol{S}^*$ in non-degenerate cases, and choosing a uniform split in degenerate cases. The proof is in Appendix B.

**Theorem 4.1.** *Define $D \subseteq \mathcal{V}^n$ to be the set of $\bar{i} \in \mathcal{V}^n$ where all $S^*_{i,\bar{i}} = 0$ in Equation (2). Then*

$$\beta(i|\bar{i}) = \begin{cases} \dfrac{S^*_{i,\bar{i}}}{\sum_{j\in\mathcal{V}} S^*_{j,\bar{i}}}, & \bar{i} \notin D, \\ \dfrac{\mathbf{1}[i \in set(\bar{i})]}{|set(\bar{i})|}, & \bar{i} \in D, \end{cases} \tag{5}$$

*solve the importance sampling optimization problem in canonical decomposition.*

Thus, canonical decomposition provides no computational advantage over the relaxed OTLP.

## 5 OPTIMAL MULTI-DRAFT SPECULATIVE SAMPLING BY MAX-FLOW

Much of the work by Hu et al. (2025) employs subset selection (Equation (4)) to efficiently compute the optimal acceptance $\alpha^* = 1 + \psi(H^*)$ for some $p_{\text{draft}}$ schemes. However, they do not have a method for computing the OT efficiently from $H^*$. They only solve for the OT when $p_{\text{draft}}$ is *greedy* (choose the top $n-1$ tokens in a draft model $q$ and sample the last without replacement), by directly solving the OTLP [1]. Excluding this $p_{\text{draft}}$, there is no known method which computes the OT for $n \geq 2$ and is faster than directly solving the OTLP or relaxed OTLP with an LP solver; this includes canonical decomposition, since Theorem 4.1 shows it reduces to the relaxed OTLP.

In this section, we formulate a more efficient method to solve the OT, using $H^*$. We first enumerate cases where efficient computation of $H^*$ is possible, building on Hu et al. (2025). Then, we show how solving for $\boldsymbol{S}^*$ in the relaxed OTLP is a max-flow problem. Finally, we reverse the subset selection derivation using complementary slackness, to obtain a more compact *feasibility* max-flow problem that depends on $H^*$. This allows simplification of Equation (3) to a sparser form.

### 5.1 EFFICIENT RECOVERY OF $H^*$

Computing $\alpha^*$ reduces to minimizing the set function $\psi : 2^{\mathcal{V}} \to \mathbb{R}$ defined in Equation (4). While this is NP-hard for general $\psi$, there are polynomial-time approaches (Iwata, 2008; Orlin, 2009) when $\psi$ is **submodular**. In their work, Hu et al. (2025) show submodularity holds if $p_{\text{draft}}$ samples i.i.d. $n$ times from a single draft distribution $q$. In Appendix C, we generalize their argument to $p_{\text{draft}}$ that samples independently from *any* $n$ draft distributions. These polynomial-time algorithms are often impractical for large $V$, but the use of top-$p/k$ sampling on the draft model(s) can restrict the domain $\mathcal{V}$ and make this feasible. Also, when $n = 2$, minimizing $\psi$ is *submodular* quadratic psuedo-boolean optimization (QBPO), where far more efficient algorithms exist: see Appendix D for more details.

When $p_{\text{draft}}$ samples i.i.d. from $q$, Hu et al. (2025) are able to improve on standard submodular algorithms to obtain an $O(V \log V)$ approach. They show that it suffices to sort $i \in \mathcal{V}$ in decreasing order of $q(i)/p(i)$, and select the *prefix* of this list which minimizes $\psi$. We will mainly be concerned with this setting, rather than general submodular $\psi$ setting, which we leave to future work.

### 5.2 MAX-FLOW REDUCTION OF RELAXED OTLP

We now describe the relaxed OTLP (Equation (2)) as a max-flow problem. Max-flow solvers are much faster than general LP solvers here, as we empirically demonstrate in Section 7.

Define a bipartite network $\Omega$ with source $s$, left vertices $\mathcal{V}$, right vertices $\mathcal{V}^n$, and sink $t$. We draw edges $s \to i$ with capacity $p(i)$ for $i \in \mathcal{V}$, edges $i \to \bar{i}$ with capacity $\infty$ for $i \in \mathcal{V}, \bar{i} \in A_i$, and edges $\bar{i} \to t$ with capacity $p_{\text{draft}}(\bar{i})$ for $\bar{i} \in \mathcal{V}^n$. Considering variables $S_{i,\bar{i}}$ as flows along $i \to \bar{i}$, we see the capacity constraints exactly reduce to the inequalities in the relaxed OTLP, and the maximization objective is the same as maximizing the flow. The zero equality constraints are handled by the fact that we only draw edges $i \to \bar{i}$ if $\bar{i} \in A_i$. Thus, $\boldsymbol{S}^*$ can be computed by running max-flow on $\Omega$.

---

[1] Here, only $O(V)$ possible draft $n$-tuples can be sampled, rather than $V^n$, so the OTLP is feasible to solve.

## 5.3 Optimizing Max-Flow with Complementary Slackness

While the max-flow formulation is advantageous over a general LP solver, it can still be prohibitive for large $\mathcal{V}^n$. Is there any way to use information like $H^*$ to reduce the size of the network $\Omega$? We find the answer is affirmative through complementary slackness, a method to reverse the dualization step in Section 3.2. For background on the dualization and the proof of below, see Appendix E.

**Theorem 5.1.** *Any nonnegative $\boldsymbol{S}^*$ defined over $\mathcal{V} \times \mathcal{V}^n$ satisfies the following system,*

$$\sum_{\bar{i} \notin (H^*)^n} S^*_{i,\bar{i}} \leq p(i) \quad \forall i \notin H^*, \quad \sum_{i \notin H^*} S^*_{i,\bar{i}} = p_{draft}(\bar{i}) \quad \forall \bar{i} \notin (H^*)^n, \tag{6a}$$

$$\sum_{\bar{i} \in (H^*)^n} S^*_{i,\bar{i}} = p(i) \quad \forall i \in H^*, \quad \sum_{i \in H^*} S^*_{i,\bar{i}} \leq p_{draft}(\bar{i}) \quad \forall \bar{i} \in (H^*)^n, \tag{6b}$$

$$S^*_{i,\bar{i}} = 0 \quad \forall i \in H^*, \bar{i} \notin (H^*)^n, \qquad S^*_{i,\bar{i}} = 0 \quad \forall i \in \mathcal{V}, \bar{i} \notin A_i, \tag{6c}$$

*if and only if it is an optimal solution to the relaxed OTLP.*

Once we eliminate all zero variables in Equation (6c) from the top two lines, we call Equation (6a) the **outer system** and Equation (6b) the **inner system**. These are independent, as the former has terms with $i \notin H^*, \bar{i} \notin (H^*)^n, \bar{i} \in A_i$ and the latter has terms with $i \in H^*, \bar{i} \in (H^*)^n, \bar{i} \in A_i$. In the same way as the relaxed OTLP, these represent max-flow problems over a bipartite network: see Appendix F for a proof of this equivalence. Thus, we can solve the outer and inner systems by creating corresponding flow networks, solving for the max-flow, and then recovering the variables along the $\infty$-capacity edges. To form the flow network for the outer system, we restrict $\Omega$ from Section 5.2 to left vertices $V \setminus H^*$ and right vertices $V^n \setminus (H^*)^n$. For the inner system, we restrict $\Omega$ to left vertices $H^*$ and right vertices $(H^*)^n$. The complexity of these networks depends on $H^*$.

Finally, to compute $\boldsymbol{C}^*$ from the full $\boldsymbol{S}^*$ given by the outer and inner system solutions, we use Equation (3). In fact, the equality conditions in Theorem 5.1 tell us some residuals are zero:

$$p^{\text{res}}(i) = 0 \quad \forall i \in H^*, \quad p^{\text{res}}_{\text{draft}}(\bar{i}) = 0 \quad \forall \bar{i} \notin (H^*)^n, \tag{7}$$

Thus, when we sample the $n$-tuple of draft tokens $\omega \in \mathcal{V}^n$, and aim to compute the slice of the OT $\pi(\cdot|\omega)$ along this $n$-tuple, we only need to solve for the variables $C^*_{i,\omega}$ for $i \in \mathcal{V}$, which simplify to

$$C^*_{i,\omega} = \begin{cases} S^*_{i,\omega}, & i \in \text{set}(\omega), \\ \dfrac{p^{\text{res}}_{\text{draft}}(\omega)}{\sum_{j \in \mathcal{V}} p^{\text{res}}(j)} \cdot p^{\text{res}}(i), & i \notin H^*, \omega \in (H^*)^n, \\ 0, & \text{else}, \end{cases} \tag{8}$$

by application of Equation (3). In fact, by independence of the outer and inner systems, examining the variables above, **we only need to solve the outer system if** $\omega \notin (H^*)^n$. When $H^*$ is large, this is a significant speedup, as the outer system network is far smaller than the one in Section 5.2.

## 6 Near-Linear Sampling via Global Resolution

Our max-flow reduction of complementary slackness shows that for arbitrary $p_{\text{draft}}$, we should not expect to be able to compute $\boldsymbol{C}^*$ in less time than the max-flow approach. Across all flow networks we constructed, the capacities were $p(i), p_{\text{draft}}(\bar{i})$ for $i \in \mathcal{V}, \bar{i} \in \mathcal{V}^n$, which are essentially free parameters. Thus, a faster algorithm would likely require a faster general max-flow solver.

However, when $p_{\text{draft}}$ is formed by i.i.d. sampling from a draft model $q$, we can exploit additional structure to solve the complementary slackness system efficiently. First, we show how to compute feasible values of the outer residuals $p^{\text{res}}(i)$ for $i \in \mathcal{V} \setminus H^*$, using polymatroid theory. This turns all conditions in Theorem 5.1 involving $p(i)$ into equalities. Once these are equalities, there exists a solution to the outer and inner systems parametrized by variables $\alpha_i$ for $i \in \mathcal{V} \setminus H^*$ and $i \in H^*$, such that each row of the solution is a softmax over a slice of variables. These variables can be computed by minimizing associated convex functions. We call this approach **global resolution**. While this is not efficient for large $\mathcal{V}$, by truncating the convex functions to a smaller subset of variables, we can solve an approximate optimization problem with negligible accuracy loss.

Global resolution (Algorithm 1) follows three key steps. For discussion on non-i.i.d. extensions of these steps, such as to distinct independent drafts or sampling without replacement, see Appendix S.

1. We solve for $H^* \subseteq \mathcal{V}$. As per Section 5.1, existing methods (Hu et al., 2025) allow recovery of $H^*$ in $O(V \log V)$ time, so we take access to $H^*$ for granted throughout this section.

2. With $H^*$, we solve for the outer residuals $p^{\text{res}}(i)$ for $i \in \mathcal{V} \setminus H^*$ in the outer system. This can be done in $O(V \log V)$ time with polymatroid theory, as in Section 6.1. This allows us to replace the $p(i)$ upper bounds in Equation (37a) with equalities to $p_i = p(i) - p^{\text{res}}(i)$.

3. We solve *truncated* convex minimization problems in Theorem 6.4 (requires $p_i$ from above) and Theorem 6.5 to find approximate global solutions to the outer and inner systems[2]. These problems are defined for general truncated variables sets $T$ in Section 6.2. In Section 6.3, we show how to carefully choose $T$, and then compute the OT from the convex solutions, to get strong provable approximation guarantees. We also show in Appendix K how to keep the convex problems under $O(|T|^n/n!)$ variables and generally speed up minimization.

Prior to our work, progress had only been made on step 1. Knowledge of only $H^*$ allows computation of the optimal acceptance $\alpha^* = \psi(H^*)$, but not the optimal verification algorithm itself.

Excluding step 3, our algorithm is $O(V \log V)$. The runtime of step 3 is quite variable, and can depend significantly on the $p$ and $q$ distributions. In some cases, the minimum truncation size $|T|$ required to achieve a good provable approximation is too large to run on the order of milliseconds, so we terminate global resolution early. Nevertheless, in practice (Table 1), this is not prohibitive.

## 6.1 OUTER RESIDUAL LP

From Equation (7), there are two classes of nontrivial residuals: the outer residuals $p^{\text{res}}(i)$ for $i \in \mathcal{V} \setminus H^*$, and the inner residuals $p^{\text{res}}_{\text{draft}}(\bar{i})$ for $\bar{i} \in (H^*)^n$. While solving for the inner residuals is difficult[3], we can compute outer residuals with the **outer residual LP** (Theorem 6.1, see Appendix G).

**Theorem 6.1** (Outer Residuals). *We have $0 \leq p_i \leq p(i)$ for $i \in \mathcal{V} \setminus H^*$ are feasible in*

$$\sum_{i \in S} p_i \leq \sum_{i \in S} p(i) + \psi(\mathcal{V} \setminus S) \quad \forall S \subseteq \mathcal{V} \setminus H^*, \tag{9}$$

*with equality at $S = \mathcal{V} \setminus H^*$, if and only if $p^{res}(i) = p(i) - p_i$ are feasible in the outer system.*

This LP has up to $2^V$ constraints, so it is infeasible to even write down. Fortunately, most of these constraints are redundant when $\psi$ is submodular. By applying polymatroid theory (see Appendix H), we get a closed-form solution by taking consecutive differences of minimums of $\psi$.

**Theorem 6.2.** *Suppose $\psi$ is submodular. Fix any ordering $\{v_1, \ldots, v_k\}$ of $\mathcal{V} \setminus H^*$. Then*

$$p_{v_i} = p(v_i) + \min_{T \supseteq H_{i+1}} \psi(T) - \min_{T \supseteq H_i} \psi(T) \quad \forall i \in [k]. \tag{10}$$

*is a solution to Theorem 6.1, where $H_i = H^* \cup \{v_i, \ldots, v_k\}$ for each $i \in [k]$.*

Computing the $\min \psi$ terms is extremely prohibitive, since there are $O(V)$ of them, so we could require up to $V$ submodular minimization calls. However, for the i.i.d. $p_{\text{draft}}$ construction, each call can be made $O(V \log V)$ with a procedure that evaluates $\psi$ across prefixes of a sorted list, by following the $q$-convexity approach in Section 4.2 of Hu et al. (2025) verbatim.

**Lemma 6.3.** *Suppose $p_{draft}$ is formed by sampling i.i.d. from $q$. Fix subsets $A \subseteq B \subseteq \mathcal{V}$. Then*

$$\min_{A \subseteq S \subseteq B} \psi(S) = \min_{0 \leq i \leq |B \setminus A|} \psi(L_i \cup A), \tag{11}$$

*where $L_i$ is the $i$th prefix of $B \setminus A$ when its elements $x$ are sorted by decreasing $q(x)/p(x)$.*

We can further optimize this to a *total* runtime of $O(V \log V)$ across all calls, by exploiting the fact that the ordering of $\mathcal{V} \setminus H^*$ in Theorem 6.2 is arbitrary. By selecting $v_1, \ldots, v_k$ in *increasing* order of $q(\cdot)/p(\cdot)$, we know from Lemma 6.3 that each $\min_{T \supseteq H_i} \psi(T)$ lies among $\psi(H_i), \ldots, \psi(H_1)$. Thus, using cumulative minimums, we can get all $\min \psi$ terms in $O(V \log V)$.

---

[2]As mentioned at the end of Section 5.3, we only need to solve the outer system if $\omega \notin (H^*)^n$.

[3]This is primarily due to the exponential number of inner residuals. Even if $p_{\text{draft}}$ is i.i.d., it is not guaranteed that some $p^{\text{res}}_{\text{draft}}$ admits a similar i.i.d. structure, without additional assumptions on $p$ and $q$.

## 6.2 Truncated Outer and Inner Convex Solvers

Once we have computed feasible values for the outer system residuals $p^{\text{res}}(i) = p(i) - p_i$ for $i \in \mathcal{V} \setminus H^*$ (Theorem 6.2), we can turn all $p(i)$ upper bounds in the outer system (Equation (6a)) into $p_i$ equality bounds. Our key observation is that as the outer system is feasible, it has some **global low-dimensional solution** parametrized over $\mathcal{V} \setminus H^*$. The rows of this solution are formed by taking the softmax of slices of the parameters, and the parameter values can be computed to arbitrary accuracy by minimizing a truncated convex function as in Theorem 6.4 (proof in Appendix I),

**Theorem 6.4** (Outer Convex Solver). *Fix $\epsilon > 0$ and $T \subseteq \mathcal{V} \setminus H^*$. Define $\Phi_T : \mathbb{R}^{\mathcal{V} \setminus H^*} \to \mathbb{R}$ by:*

$$\Phi_T((\alpha_i)_{i \in \mathcal{V} \setminus H^*}) = \sum_{\bar{i} \in (H^* \cup T)^n \setminus (H^*)^n} p_{draft}(\bar{i}) \log \left( \sum_{i \in set(\bar{i}) \setminus H^*} e^{\alpha_i} \right) - \sum_{i \in T} p_i \alpha_i. \tag{12}$$

*This is constant over $\alpha_i$ for $i \notin T$. Then there is some $(\alpha_i^*)_{i \in \mathcal{V} \setminus H^*}$ such that*

$$\|\nabla \Phi_T((\alpha_i^*)_{i \in \mathcal{V} \setminus H^*})\|_1 \leq \epsilon + \epsilon_T, \quad \epsilon_T = 1 - \left( \sum_{i \in H^* \cup T} q(i) \right)^n. \tag{13}$$

*Furthermore, for any $(\alpha_i)_{i \in \mathcal{V} \setminus H^*}$, the following variables solve the outer system,*

$$S_{i,\bar{i}} = \frac{e^{\alpha_i}}{\sum_{j \in set(\bar{i}) \setminus H^*} e^{\alpha_j}} \cdot p_{draft}(\bar{i}) \quad \forall i \in set(\bar{i}) \setminus H^*, \bar{i} \in \mathcal{V}^n \setminus (H^*)^n, \tag{14}$$

*with at most $\|\nabla \Phi_T((\alpha_i)_{i \in \mathcal{V} \setminus H^*})\|_1 + 3\epsilon_T$ total L1 deviation from the $p(i)$ equality constraints.*

The inner system can be approached similarly, but the function now includes a slack term[4], as we have not solved for the inner residuals $p_{\text{draft}}^{\text{res}}(\bar{i})$. The proof can be found in Appendix J.

**Theorem 6.5** (Inner Convex Solver). *Fix $\epsilon > 0$ and $T \subseteq H^*$. Define $\Theta_T : \mathbb{R}^{H^*} \to \mathbb{R}$ by:*

$$\Theta_T((\alpha_i)_{i \in H^*}) = \sum_{\bar{i} \in T^n} p_{draft}(\bar{i}) \log \left( 1 + \sum_{i \in set(\bar{i})} e^{\alpha_i} \right) - \sum_{i \in T} p(i) \alpha_i. \tag{15}$$

*This is constant over $\alpha_i$ for $i \in H^* \setminus T$. Then there is some $(\alpha_i^*)_{i \in H^*}$ such that*

$$\|\nabla \Theta_T((\alpha_i)_{i \in H^*})\|_1 \leq \epsilon + \gamma_T, \quad \gamma_T = \left( \sum_{i \in H^*} q(i) \right)^n - \left( \sum_{i \in T} q(i) \right)^n. \tag{16}$$

*Furthermore, for any $(\alpha_i)_{i \in H^*}$, the following variables solve the inner system,*

$$S_{i,\bar{i}} = \frac{e^{\alpha_i}}{1 + \sum_{j \in set(\bar{i})} e^{\alpha_j}} \cdot p_{draft}(\bar{i}) \quad \forall i \in set(\bar{i}), \bar{i} \in (H^*)^n, \tag{17}$$

*with at most $\|\nabla \Theta_T((\alpha_i)_{i \in H^*})\|_1 + 3\gamma_T$ total L1 deviation from the $p(i)$ equality constraints.*

In essence, solving for a global softmax solution in complementary slackness reduces to minimizing (finding a point with near-zero gradient) a convex function defined over variables in the truncated set $T$: this is $\Phi_T$ for the outer system and $\Theta_T$ for the inner system. When minimizing $\Phi_T$ and $\Theta_T$, the total deviation from the $p(i)$ equality constraints is guaranteed to fall under $5\epsilon_T$ and $5\gamma_T$, respectively, by setting $\epsilon = \epsilon_T, \gamma_T$ in the solvers and using the gradient existence statements from the theorems.

When we increase the size of $T$, the error bounds $\epsilon_T$ and $\gamma_T$ (the **tunable errors**) approach zero. Thus, we can approximate the true complementary slackness solution to arbitrary accuracy. To prevent excessive runtime, we may **terminate early** if the size $|T|$ of the convex minimization problem exceeds a preset threshold, or a maximum iteration count in the minimization algorithm is reached. Due to space constraints, we clarify these details and our minimization approach in Appendix K.

---

[4]We cannot do this for the outer system. This solver has one variable for each equality constraint. However, the outer system has exponentially many ($|\mathcal{V}^n \setminus (H^*)^n|$) equalities before outer residual computation. Thus, the polymatroid approach is necessary to make even a truncated convex solver practical for the outer system.

---

**Algorithm 1:** Global Resolution

---

**Input:** $\mathcal{V}, p, q, n$, error threshold $\tau$, draft count $n$, drafted $n$-tuple $\omega = (\omega_1, \ldots, \omega_n) \sim q(\cdot)$
**Output:** Optimal transport slice $\pi(\cdot|\omega)$

1 Compute $H^*$ from $p, q, \mathcal{V}, n$ as in Section 5.1
2 Compute $p_i$ for all $i \in \mathcal{V} \setminus H^*$ from $p, q, \mathcal{V}, n$ as in Theorem 6.2 and Lemma 6.3
3 **if** $\omega_1, \ldots, \omega_n \in H^*$ **then**
4      Choose minimal $T \subseteq H^*$ such that $\gamma_T \leq \tau$
5      Find inner system variables $S_{i,\bar{i}}$ using Theorem 6.5, stop at $\|\nabla \Theta_T\|_1 \leq 5\tau$
6      **return** normalized $C_{\cdot,\omega}$ from $S_{i,\bar{i}}$ and $p_i$ in Lemma 6.6
7 Choose minimal $T \subseteq H^*$ such that $\epsilon_T \leq \tau$
8 Find outer system variables $S_{i,\bar{i}}$ using Theorem 6.4, stop at $\|\nabla \Phi_T\|_1 \leq 5\tau$
9 **return** normalized $C_{\cdot,\omega}$ from $S_{i,\bar{i}}$ in Lemma 6.6

---

## 6.3 TRUNCATION SELECTION AND APPROXIMATION GUARANTEES

Once we have used the outer and inner convex solvers to compute the approximation $S$ to the real complementary slackness solution $S^*$ (all variables not in these systems are zero), we plug in the former into Equation (3) to obtain an approximation $C$ to the real optimal transport solution $C^*$:

$$C_{i,\bar{i}} = \begin{cases} S_{i,\bar{i}}, & i \in \text{set}(\bar{i}), \\ \dfrac{p(i) - p_i}{\sum_{j \in \mathcal{V} \setminus H^*} (p(j) - p_j)} \cdot \left( p_{\text{draft}}(\bar{i}) - \sum_{i \in \text{set}(\bar{i})} S_{i,\bar{i}} \right), & i \notin H^*, \bar{i} \in (H^*)^n, \\ 0, & \text{else}, \end{cases} \quad (18)$$

The following lemma (proof in Appendix L) ensures that the tunable errors in Theorem 6.4 and Theorem 6.5 translate directly to bounds on deviations in the OTLP solution and acceptance rates.

**Lemma 6.6** (Approximation Guarantee). *Solve for $p_i$ over $i \in \mathcal{V} \setminus H^*$ as in Theorem 6.2. Using these values, let $S_{i,\bar{i}}$ be defined as in Theorem 6.4 with L11 deviation $\alpha$ and Theorem 6.5 with L1 deviation $\beta$. Then $C_{i,\bar{i}}$ as defined in Equation (18) satisfies the OTLP, with up to $\alpha + 2\beta$ total L1 deviation in the $p(i)$ equality constraints and $\alpha + \beta$ total deviation in the optimal acceptance rate.*

Using the guaranteed $5\epsilon_T$ and $5\gamma_T$ deviations for the outer and inner solvers, we can therefore choose suitable $T$ to ensure the tunable errors $\epsilon_T, \gamma_T$ do not exceed a specified error threshold $\tau$, as Lemma 6.6 ensures that sampling from the OT induced by this approximate $C$ will deviate by at most $15\tau$ from the true target distribution in L1 distance, and deviate from the optimal acceptance rate by at most $10\tau$. We present the full global resolution algorithm with the error threshold in Algorithm 1.

## 7 EXPERIMENTS

In this section we empirically test the efficacy of our new algorithms for i.i.d. multi-draft, single-step setting. First, we show that optimal acceptance rates can substantially increase with a higher number of drafts $n$, and larger $k$ in top-$k$ sampling of the draft model. We then compare our algorithms' solve times against a standard LP solver for various $k, n$. Our algorithms are practical for a much wider range of $k, n$, achieving state-of-the-art optimal acceptance rates in multi-draft speculative sampling. Details on our setup for both acceptance and solve time experiments are given in Appendix M.

### 7.1 OPTIMAL ACCEPTANCE FOR INCREASING $k$ AND $n$

For our target and draft models, we use the pairs Gemma-2 27B/2B and Llama-3 70B/8B (Touvron et al., 2023a; Team, 2024). We test $n \leq 10$ and $k \in \{10, 100, \ldots, 10^{\lfloor \log V \rfloor}, V\}$. Note that top-$k$ sampling truncates the vocabulary $\mathcal{V}$ to size $k$ in the **relaxed OTLP**, so that compute-heavy baseline approaches like LP and max-flow solvers remain feasible. For more detail, see Appendix N.

Our results are shown in Figure 1. We observe significant improvements in acceptance as $k$ increases from 10 to 1000: for example, a marginal increase from 0.8 to 0.95 means nearly 10% more tokens

are generated at no extra cost. However, there are diminishing returns for $k \geq 10000$. Thus, top-$k$ sampling is a viable method to reduce OTLP complexity without sacrificing acceptance. There is also a steady improvement from increasing $n$ for larger $k$, demonstrating the utility of higher draft counts.

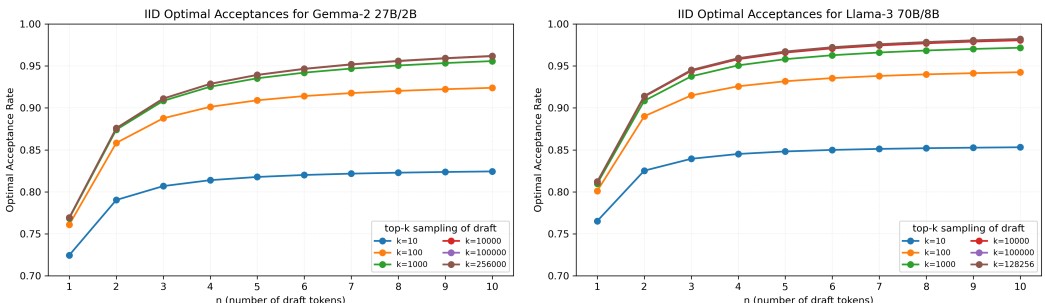

Figure 1: Optimal acceptance rates from $n$ i.i.d drafts with top-$k$ sampling with $n$ with target/draft pairs of Gemma-2 27B/2B and Llama-3 70/8B. Increasing $k$ improves acceptance rate significantly up to $k = 1000$, and increasing $n$ also results in steady increase in optimal acceptance.

Further, in Appendix O, we compare these acceptance rates to those for the greedy construction by Hu et al. (2025), described at the start of Section 5. For both Gemma-2 and Llama-3, we find that i.i.d. acceptance rates are higher than greedy for $k \geq 100$, with improvements near $2\%$ for larger $k$ and $n$.

| $(k, n)$ | **General LP** | **Max-Flow** | **Opt. Max-Flow** | **G.R. ($\tau = 10^{-3}$)** | **G.R. ($\tau = 10^{-4}$)** |
|---|---|---|---|---|---|
| $(10, 2)$ | 4.07 | **2.45** | 2.51 | 5.27 (98%) | 7.62 (93%) |
| $(10, 3)$ | 30.14 | **3.80** | 3.88 | 17.35 (98%) | 23.95 (87%) |
| $(10, 4)$ | 4000+ | 74.21 | 74.37 | **40.30 (97%)** | 54.84 (86%) |
| $(10, 5)$ | 400000+ | 10000+ | 10000+ | **70.75 (96%)** | 94.79 (85%) |
| $(100, 2)$ | 4000+ | 72.28 | 63.03 | **23.92 (38%)** | 25.47 (23%) |
| $(100, 3)$ | 400000+ | 200000+ | 200000+ | **30.52 (23%)** | 38.44 (14%) |
| $(1000, 2)$ | OOM | 400000+ | 400000+ | **34.94 (31%)** | 47.46 (15%) |

Table 1: Average Llama-3 solve times (ms/token) over $k, n$, for the five i.i.d. OTLP solvers. General LP and max-flow are baselines, and optimized max-flow and global resolution ($\tau = 10^{-3}, 10^{-4}$) are ours. Lower numbers are better. Red numbers are lower bounds from small scale tests due to excessive runtime. Global resolution can be 10,000+ times faster than others. Global resolution deviates from the target distribution by at most $15\tau$ in L1 distance, and from optimal acceptance by at most $10\tau$. We include success rates for global resolution as it can terminate early sometimes.

### 7.2 OTLP SOLVE TIME EXPERIMENTS

Here, we compute empirical average solve times (per token) for the *relaxed OTLP* across 40 random prompts from each dataset, for five methods: **(1)** a general LP solver, **(2)** a max-flow solver (Section 5.2), **(3)** an optimized max-flow solver (Section 5.3, but using normal max-flow when solving both inner and outer systems for stability), **(4)** global resolution (Section 6) with $\tau = 0.001$, and **(5)** global resolution with $\tau = 0.0001$. Lemma 6.6 ensures **(4)** and **(5)** deviate from the target distribution by at most 0.015 and 0.0015 in L1 distance, and are within 0.01 and 0.001 of optimal acceptance.

Our compared average solve times for Llama-3 are shown in Table 1. For global resolution, we also include solve success rates, since it can terminate early. We only include $(k, n)$ pairs where at least one of the first three solvers finishes in under ten minutes per token. We see that the global resolution solver is sometimes nearly ten thousand times faster than the other three methods, with runtimes always under 100 ms/token. In Appendix P, Gemma-2 results show a similar trend.

Finally, in Table 2, we enforce average time limits of 10 and 100 ms/token, and compute the acceptance for each solver's best $(k, n)$ setting that adheres to the limit. Since global resolution can fail, we default to the best of solvers **(1)**, **(2)**, **(3)** in this case, and only keep settings whose aggregate

average runtime falls under the time limit. For Llama-3, global resolution with $\tau = 0.001$ and $\tau = 0.0001$ achieve $1.03\%$ and $0.47\%$ higher acceptances than the other solvers for 100 ms/token, and $1.71\%$ and $1.16\%$ higher acceptances for 10 ms/token. While the improvements for Gemma-2 are smaller, they are significant. Thus, global resolution is the state-of-the-art OTLP solver.

| Limit | Llama-3 | | | | Gemma-2 | | | |
|---|---|---|---|---|---|---|---|---|
| (ms/tok) | **Gen. LP** | **M.F.** | **G.R.** $\tau = 10^{-3}$ | **G.R.** $\tau = 10^{-4}$ | **Gen. LP** | **M.F** | **G.R.** $\tau = 10^{-3}$ | **G.R.** $\tau = 10^{-4}$ |
| 10 | 82.53% | 83.94% | **85.65%** | 85.10% | 79.03% | 80.69% | **81.90%** | 81.35% |
| 100 | 83.94% | 89.01% | **90.04%** | 89.46% | 80.69% | 85.83% | **86.60%** | 86.07% |

Table 2: Best acceptance rates for baseline solvers (general LP) and ours (max-flow, optimized max-flow, global resolution with $\tau = 10^{-3}, 10^{-4}$) on Llama-3 and Gemma-2 under average solve time limits of 10 and 100 ms/token. Optimized max-flow numbers are the same as max-flow.

### 7.3 TEMPERATURE EXPERIMENTS

All results above are for the temperature 1 sampling of the target model. In Appendix Q, we extend our acceptance rate analysis to temperature $0.2, 0.4, 0.6, 0.8$ target sampling on Gemma-27B/2B. We find that for temperatures below $0.8$, increasing $k$ from 10 rarely improves the optimal acceptance rate. This has the implication that approaches other than global resolution (general LP, max-flow, optimized max-flow) can be efficient and achieve high acceptance in low temperature settings.

In Appendix Q, we also plot the solve time and success rate of global resolution with $\tau = 10^{-4}$ as temperature varies. We find that global resolution has significantly lower failure rates and solve times at low temperatures, because truncation can both be efficient and give a robust approximation for more concentrated distributions. In other words, our results use the worst setting for global resolution.

### 7.4 MULTI-STEP EXPERIMENTS

We extend our method to the multi-step framework SpecTr (Sun et al., 2023) to evaluate the real-world latency of global resolution. Our results are shown in Appendix R, alongside theoretical explanations of how global resolution errors scale with draft block length in the multi-step setting. We see that global resolution is highly effective when integrated into multi-step frameworks, improving walltime decoding by nearly $2\times$ compared to the baseline, even when using a naive OTLP heuristic in failure cases. By using better OTLP solvers, such as K-SEQ (Sun et al., 2023), in such failure cases, we expect further decoding improvements to materialize.

## 8 CONCLUSION

Building on previous theoretical work in multi-draft speculative sampling, we derived a complementary slackness system to solve for the optimal transport (OT) parameters. We showed how this could speed up max-flow OT solvers through an approach called global resolution. We empirically demonstrated our algorithm achieved high acceptance rates with little overhead. Future work could examine the efficacy of this algorithm in the multi-step setting, or extend it to non-i.i.d. draft settings.

### REPRODUCIBILITY STATEMENT

The majority of work is theoretical, and can be verified through the in-depth proofs we have provided in Appendices B to J. While the details behind our convex minimization approach are not included in the main body due to space constraints, we provide all necessary information to reproduce the gradient descent approach for our convex solvers in Appendix K. This includes our simplification of the functions to minimize, how to efficiently compute gradient values, what open-source minimizer to use, what thresholds are set, and how many iterations to run. We also describe our precise machine setup, dataset composition, and LP and max-flow solvers for baseline approaches in Appendix M, and we describe our experimental approach in Section 7.

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

## A    REVIEW OF SINGLE-STEP SPECULATIVE SAMPLING

Single-draft speculative sampling aims to accelerate the autoregressive decoding of an expensive target model $M_p$ with a cheaper draft model $M_q$, which share a common vocabulary $\mathcal{V}$ of size $V$. In autoregressive decoding, the token context $c$ induces target and draft distributions $p(x|c), q(y|c)$. Note that top-$p/k$ and temperature sampling simply change how $p, q$ are computed from the logits $M_p(c), M_q(c)$, so these can be plugged into any speculative sampling method.

Now, a joint coupling $\pi(x, y)$ of $p, q$ is any joint distribution over $\mathcal{V} \times \mathcal{V}$ with marginal distributions $p, q$. Each coupling induces a transport $\pi(x|y)$ from $q$ to $p$, which is a conditional distribution where sampling $y_0 \sim q(y)$ and $x_0 \sim \pi(x|y_0)$ is equivalent to sampling $x_0 \sim p(x)$. In speculative sampling, one samples a candidate token from the draft distribution $q$ and passes it through the transport induced by some valid coupling $\pi$, to ensure the output matches the target distribution $p$.

The full *single-step* procedure is as follows. The key assumption which allows speculative sampling to improve latency is that computing $p(x|c), p(x|c, y_0)$ in parallel is not much slower than computing $p(x|c)$ alone, since LLM inference is often memory-bound rather than compute-bound.

1. Given a context $c$, sample $y_0 \sim q(y|c)$ from the draft model.
2. Compute target probabilities $p(x|c), p(x|c, y_0)$ in parallel.
3. Sample $x_0 \sim \pi(x|y_0)$ for some valid transport $\pi$, and append it to the context $c$.
4. If $x_0 = y_0$, sample a token from $p(y|c, x_0)$ and append it to the context $c$.

Thus, whenever $x_0 = y_0$ above, we generate *two* tokens (steps 3 and 4) at the cost of essentially one target forward pass, obtaining nearly a $2\times$ speedup. This means to maximize the speedup and achieve *optimal* single-draft speculative sampling, one must maximize $\mathbb{P}(x_0 = y_0) = \sum_{x \in \mathcal{V}} \pi(x, x)$, achieved for the *optimal transport* $\pi^*$. This is called the *acceptance rate*, the probability that the $q$-sampled token is not altered by the transport. In the original speculative sampling work, Chen et al. (2023); Leviathan et al. (2023) computed a closed-form expression for $\pi^*$ based on $p(x|c), q(x|c)$.

Sun et al. (2023) extended this procedure to single-step *multi-draft*, by replacing $q(x|c)$ with a general $n$-token distribution $p_{\text{draft}}(\boldsymbol{y}|c)$ over $\boldsymbol{y} \in \mathcal{V}^n$, which can be formed by i.i.d. sampling one draft model, independently sampling different draft models, or a variety of other methods. Now, a joint coupling $\pi(x, \boldsymbol{y})$ of $p, p_{\text{draft}}$ is a joint distribution over $\mathcal{V} \times \mathcal{V}^n$ with marginals $p, p_{\text{draft}}$, which induces a transport $\pi(x|\boldsymbol{y})$ from $p_{\text{draft}}$ to $p$. Here, the key assumption we make is slightly stronger: that computing $p(x|c), p(x|c, y_0), \ldots, p(x|c, y_{n-1})$ is parallel is not much slower than computing $p(x|c)$ alone. This is generally true as long as $n$ is not too large. The procedure is:

1. Given a context $c$, sample $(y_0, \ldots, y_{n-1}) \sim p_{\text{draft}}(\boldsymbol{y}|c)$ from the draft scheme.
2. Compute target probabilities $p(x|c), p(x|c, y_0), \ldots, p(x|c, y_{n-1})$ in parallel.
3. Sample $x_0 \sim \pi(x|y_0, \ldots, y_{n-1})$ for some valid transport $\pi$, and append it to the context $c$.
4. If $x_0 \in \{y_0, \ldots, y_{n-1}\}$, sample a token from $p(y|c, x_0)$ and append it to the context $c$.

Similar to the single-draft case, we now get a nearly $2\times$ speedup whenever $x_0 \in \{y_0, \ldots, y_{n-1}\}$. Thus, we want to maximize the probability that $x_0$ lies among the $n$ sampled draft tokens, which is the multi-draft *acceptance rate*. Combined with the conditions necessary for $\pi$ to be a valid joint coupling, this leads to the OTLP formulation.

## B    EQUIVALENCE OF CANONICAL DECOMPOSITION AND RELAXED LP

We first state canonical decomposition. Note that the $\beta$-optimization problem below is exponentially large and is not an LP, so it is infeasible to solve without truncating $\mathcal{V}$, as Khisti et al. (2025) do.

**Theorem B.1.** *(Khisti et al., 2025) Define variables $\beta(i|\bar{i})$ for $\bar{i} \in \mathcal{V}^n, i \in \mathcal{V}$, set to zero for $i \notin set(\bar{i})$. Then the $\beta$ which solves the following gives the optimal importance sampling parameters:*

$$\max_{\beta \succeq 0} \sum_{i \in \mathcal{V}} \min \left( p(i), \sum_{\bar{i} \in \mathcal{V}^n} \beta(i|\bar{i}) p_{draft}(\bar{i}) \right) \quad s.t. \quad \sum_{i \in \mathcal{V}} \beta(i|\bar{i}) = 1 \quad \forall \bar{i} \in \mathcal{V}^n. \quad (19)$$

*In fact, the objective value achieved by this optimal $\beta$ is the optimal acceptance rate $\alpha^*$.*

Now, we prove Theorem 4.1, by showing that our defined $S^*_{i,\bar{i}}$ satisfy the constraints above.

*Proof.* We first show these satisfy the constraints of Theorem B.1. Indeed, these are all nonnegative. Also, by the relaxed LP constraints, we have $S^*_{i,\bar{i}} = 0$ for all $\bar{i} \notin A_i$. Thus, by definition of the incidence set $A_i$, and considering the first and second cases in the $\beta$ definition, we have $\beta(i|\bar{i}) = 0$ if $i \notin \text{set}(\bar{i})$. Furthermore, for all $\bar{i} \in \mathcal{V}^n$, we see that

$$\sum_{i \in \mathcal{V}} \beta(i|\bar{i}) = \sum_{i \in \mathcal{V}} \frac{S^*_{i,\bar{i}}}{\sum_{j \in \mathcal{V}} S^*_{j,\bar{i}}} = 1 \tag{20}$$

in the case that $\bar{i} \notin D$. We also see that

$$\sum_{i \in \mathcal{V}} \beta(i|\bar{i}) = \sum_{i \in \text{set}(\bar{i})} |\text{set}(\bar{i})|^{-1} + \sum_{i \notin \text{set}(\bar{i})} 0 = 1 \tag{21}$$

if $\bar{i} \in D$. Thus, all constraints are satisfied.

Now, we show that the objective value with these $\beta$ in Theorem B.1 is at least the optimal objective value with $\boldsymbol{S}^*$ in the relaxed LP. This will complete the proof, as the optimal objective values in Theorem B.1 and the relaxed LP are both $\alpha^*$. To show this, we use the other relaxed LP constraints. By the the second inequality constraint,

$$\sum_{\bar{i} \in \mathcal{V}^n} \beta(i|\bar{i}) p_{\text{draft}}(\bar{i}) \geq \sum_{\bar{i} \notin D} \beta(i|\bar{i}) p_{\text{draft}}(\bar{i}) = \sum_{\bar{i} \notin D} S^*_{i,\bar{i}} \frac{p_{\text{draft}}(\bar{i})}{\sum_{j \in \mathcal{V}} S^*_{j,\bar{i}}} \geq \sum_{\bar{i} \notin D} S^*_{i,\bar{i}} \tag{22}$$

for all $i \in \mathcal{V}$. Furthermore, for all $\bar{i} \in \mathcal{V}^n$, the first inequality constraint in the relaxed LP shows that

$$\min\left(p(i), \sum_{\bar{i} \notin D} S^*_{i,\bar{i}}\right) = \sum_{\bar{i} \notin D} S^*_{i,\bar{i}}. \tag{23}$$

Therefore, the objective in Theorem B.1 is *at least*

$$\sum_{i \in \mathcal{V}} \sum_{\bar{i} \notin D} S^*_{i,\bar{i}} = \sum_{\bar{i} \notin D} \sum_{i \in V} S^*_{i,\bar{i}} = \sum_{\bar{i} \in \mathcal{V}^n} \sum_{i \in V} S^*_{i,\bar{i}} = \alpha^*. \tag{24}$$

This completes the proof that our defined $\beta$ are optimal. $\qquad\square$

## C    COMPUTING $H^*$ FOR INDEPENDENT DRAFTS IS SUBMODULAR MINIMIZATION

Here, we show that the subset selection problem $\alpha^* = 1 + \min_{H \subseteq \mathcal{V}} \psi(H)$ is an instance of submodular minimization when $p_{\text{draft}}$ is formed by independently sampling from $q_1, \ldots, q_n$ to get $n$ tokens. This generalizes the work of Hu et al. (2025), who proved this in the case $q_1 = \ldots = q_n$.

**Theorem C.1.** *When $p_{draft}$ independently samples from $q_1, \ldots, q_n$, the set function $\psi$ in submodular.*

*Proof.* First, we can define $\psi(H) = P(H) - Q(H)$, where

$$P(H) = \sum_{i \in H} p(i), \quad Q(H) = \sum_{\bar{i} \in H^n} p_{\text{draft}}(\bar{i}). \tag{25}$$

Then, because $p_{\text{draft}}(i_1, \ldots, i_n) = q_1(i_1) \cdots q_n(i_n)$, we have

$$Q(H) = \sum_{i_1, \ldots, i_n \in H} \prod_{j=1}^{n} q_j(i_j) = \prod_{j=1}^{n} \left(\sum_{i \in H} q_j(i)\right). \tag{26}$$

Note that $P(H_1 \cap H_2) + P(H_1 \cup H_2) = P(H_1) + P(H_2)$ for any $H_1, H_2 \subseteq \mathcal{V}$, so $P$ is a modular function. Thus, to show $\psi = P - Q$ is *submodular*, it suffices to show that $Q$ is *supermodular*.

To this end, take any $H_1, H_2 \subseteq \mathcal{V}$. We want to show $Q(H_1 \cap H_2) + Q(H_1 \cup H_2) \geq Q(H_1) + Q(H_2)$. For each $j = 1, \ldots, n$, denote $a_j = \sum_{i \in H_1} q_j(i)$, $b_j = \sum_{i \in H_2} q_j(i)$, and $c_j = \sum_{i \in H_1 \cap H_2} q_j(i)$. Then, we see $a_j + b_j - c_j = \sum_{i \in H_1 \cup H_2} q_j(i)$, and $a_j, b_j \geq c_j \geq 0$ by nonnegativity of $q_j$. The condition we want to show is equivalent to

$$\prod_{j=1}^{n}(a_j + b_j - c_j) + \prod_{j=1}^{n} c_j \geq \prod_{j=1}^{n} a_j + \prod_{j=1}^{n} b_j. \tag{27}$$

Now, define the sums $B$ and $C$ as

$$B = \sum_{A \subseteq [n], A \neq \emptyset} \left[ \prod_{i \in A}(a_j - c_j) \prod_{i \in [n] \setminus A} b_j \right], \tag{28}$$

$$C = \sum_{A \subseteq [n], A \neq \emptyset} \left[ \prod_{i \in A}(a_j - c_j) \prod_{i \in [n] \setminus A} c_j \right]. \tag{29}$$

We can see $B \geq C$ by comparing corresponding terms in the sum, since each $b_j \geq c_j \geq 0$ and each $a_j - c_j \geq 0$. Furthermore, we have the following identities by straightforward expansion of products of terms of the form $a_j - c_j + \star$:

$$\prod_{j=1}^{n}(a_j - c_j + b_j) = \prod_{j=1}^{n} b_j + B, \tag{30}$$

$$\prod_{j=1}^{n}(a_j - c_j + c_j) = \prod_{j=1}^{n} c_j + C. \tag{31}$$

Subtracting these and using $B \geq C$ gives

$$\prod_{j=1}^{n}(a_j + b_j - c_j) - \prod_{j=1}^{n} a_j \geq \prod_{j=1}^{n} b_j - \prod_{j=1}^{n} c_j, \tag{32}$$

as desired, which completes the proof. $\qquad\square$

## D   COMPUTING $H^*$ FOR $n = 2$ INDEPENDENT DRAFTS IS SUBMODULAR QPBO

Here, we describe the reduction of $H^*$ computation to submodular quadratic psuedo-boolean optimization (QPBO) when $p_{\text{draft}}$ samples independently from $n = 2$ draft models $q_1, q_2$. By writing $\psi(H)$ in the same form as in Appendix C, but now for the case $n = 2$, we see

$$\psi(H) = \sum_{i \in H} p(i) - \left( \sum_{i \in H} q_1(i) \right) \left( \sum_{i \in H} q_2(i) \right). \tag{33}$$

Denote the indicator vector $\boldsymbol{x}$ with $x_i = \mathbf{1}[i \in H]$ for each $i \in \mathcal{V}$. Then this becomes

$$\rho(\boldsymbol{x}) = \sum_{i \in \mathcal{V}} p(i) x_i + \sum_{i \in \mathcal{V}} \sum_{i \in \mathcal{V}} q_1(i) q_2(j) x_i x_j. \tag{34}$$

Minimizing $\psi(H)$ over all $H \subseteq \mathcal{V}$ is thus equivalent to minimizing $\rho(\boldsymbol{x})$ over all binary vectors $\boldsymbol{x} \in \{0, 1\}^{\mathcal{V}}$. This is precisely the form of QPBO, and we already know $\psi$ is submodular from Appendix C, so it is indeed an instance of submodular QPBO.

Our submodular QPBO instance is dense, with $n = V$ variables and $m = O(V^2)$ pairwise terms. By previous work, this reduces to a single $s - t$ min-cut computation (Kolmogorov & Zabih, 2004). Using a max-flow algorithm, the theoretical runtime is $O(m\sqrt{n}) = O(V^{2.5})$. In fact, practical implementations like the push-relabel algorithm can achieve near-quadratic runtime (Boykov & Kolmogorov, 2004). This is far better than the general submodular minimization algorithm (Orlin, 2009), which takes $O(V^5 \mathcal{E} + V^6) = O(V^6)$ time, where $\mathcal{E} = O(V)$ is the time complexity of evaluating $\psi$.

# E   COMPLEMENTARY SLACKNESS DERIVATION

We first review the details of dualization and simplification to subset selection, starting from the relaxed OTLP (Equation (2)). These details are necessary to formulate complementary slackness.

*Proof.* Dualizing the relaxed OTLP gives the **dual LP**. The zero equality constraints correspond to eliminating entries of $x$, and can be ignored. So, to transform to the dual, one introduces a variable for each inequality constraint, say $y_i$ for constraints over $i \in \mathcal{V}$, and $z_i$ for constraints over $\bar{i} \in \mathcal{V}^n$:

$$\min_{\boldsymbol{y}, \boldsymbol{z} \succeq 0} \left[ \sum_{i \in \mathcal{V}} y_i p(i) + \sum_{\bar{i} \in \mathcal{V}^n} z_{\bar{i}} p_{\text{draft}}(\bar{i}) \right] \quad \text{s.t.} \quad y_i + z_{\bar{i}} \geq 1 \quad \forall i \in \mathcal{V}, \bar{i} \in A_i. \tag{35}$$

Finally, to reduce the dual LP, Hu et al. (2025) observe that the inequality constraints form a TUM matrix. Any LP with integral inequality bounds and such constraints has an *integral* optimal solution (Hoffman & Kruskal, 2009). Also, any $y_i \geq 2$ or $z_{\bar{i}} \geq 2$ can be reduced to $1$ while upholding each constraint $y_i + z_{\bar{i}} \geq 1$ and not increasing the objective value: this means that some optimal solution is *binary*. Then, denoting $H = \{i \in \mathcal{V} : y_i = 1\}$, one can compute optimal $\boldsymbol{y}, \boldsymbol{z}$ *given fixed* $H$ as:

$$y_i = \begin{cases} 1 & i \in H, \\ 0 & i \notin H, \end{cases} \quad z_{\bar{i}} = \begin{cases} 1 & \bar{i} \notin H^n, \\ 0 & \bar{i} \in H^n. \end{cases} \tag{36}$$

Putting this back into the dual LP and simplifying gives the **subset selection** formulation. $\qquad\square$

Now, we provide a proof of Theorem 5.1, which makes use of the idea of *complementary slackness*. Primal and dual LPs have the same optimal objective value, and complementary slackness allows one to get from any optimal solution of the dual to some optimal solution of the primal. This asserts that the following two conditions are equivalent for any pair of *feasible* solutions $(\boldsymbol{\alpha}^*, \boldsymbol{\beta}^*)$ to the primal and dual LPs: **(A)** $\boldsymbol{\alpha}^*$ is optimal in the primal LP and $\boldsymbol{\beta}^*$ is optimal in the dual LP, and **(B)** each variable in $\boldsymbol{\alpha}^*, \boldsymbol{\beta}^*$ is zero, or its corresponding inequality constraint in the other LP is an equality.

In our case, considering the primal and dual LPs as the relaxed OTLP and its dual, we already have the optimal solution $\boldsymbol{\beta}^*$ to the dual from Equation (36) and the $H^*$ computation. Thus, computing an optimal primal solution $\boldsymbol{\alpha}^*$ is equivalent to jointly solving the complementary slackness constraints in **(B)** alongside the feasibility constraints for $\boldsymbol{\alpha}^*$ in the primal LP.

*Proof.* For the relaxed OTLP and its dual LP, complementary slackness constraints become:

$$y_i^* = 0 \quad \text{or} \quad \sum_{\bar{i} \in \mathcal{V}^n} S_{i,\bar{i}}^* = p(i) \qquad \forall i \in \mathcal{V}, \tag{37a}$$

$$z_{\bar{i}}^* = 0 \quad \text{or} \quad \sum_{i \in \mathcal{V}} S_{i,\bar{i}}^* = p_{\text{draft}}(\bar{i}) \qquad \forall \bar{i} \in \mathcal{V}^n, \tag{37b}$$

$$S_{i,\bar{i}}^* = 0 \quad \text{or} \quad y_i^* + z_{\bar{i}}^* = 1 \qquad \forall i \in \mathcal{V}, \bar{i} \in \mathcal{V}^n. \tag{37c}$$

Given the optimal solution $\boldsymbol{y}^*, \boldsymbol{z}^*$ to the dual LP, we only need to solve this system, alongside the original constraints in the primal (relaxed) LP, to obtain the optimal solution $\boldsymbol{S}^*$ to the primal LP. This can be done by using the binary vector expressions for $\boldsymbol{y}^*, \boldsymbol{z}^*$ in terms of $H^*$ in Equation (36). First, for Equation (37a), observe that $y_i^* \neq 0$ if and only if $i \in H^*$. Thus, it is equivalent to

$$\sum_{\bar{i} \in \mathcal{V}^n} S_{i,\bar{i}}^* = p(i) \quad \forall i \in H^*. \tag{38}$$

Next, for Equation (37b), recall that $z_{\bar{i}}^* \neq 0$ if and only if $\bar{i} \notin (H^*)^n$. So, this condition becomes

$$\sum_{i \in \mathcal{V}} S_{i,\bar{i}}^* = p_{\text{draft}}(\bar{i}) \quad \forall \bar{i} \notin (H^*)^n. \tag{39}$$

Finally, by observing the four joint cases for $y_i^*, z_{\bar{i}}^*$, we have

$$
y_i^* + z_{\bar{i}}^* = \begin{cases} 2 & i \in H^*, \bar{i} \notin (H^*)^n, \\ 1 & i \in H^*, \bar{i} \in (H^*)^n, \\ 1 & i \notin H^*, \bar{i} \notin (H^*)^n, \\ 0 & i \notin H^*, \bar{i} \in (H^*)^n, \end{cases} \tag{40}
$$

so that Equation (37c) is equivalent to

$$
\begin{aligned}
S_{i,\bar{i}} = 0 & \quad \forall i \in H^*, \bar{i} \notin (H^*)^n, \\
S_{i,\bar{i}} = 0 & \quad \forall i \notin H^*, \bar{i} \in (H^*)^n.
\end{aligned} \tag{41}
$$

The second of these can be eliminated, as it is redundant with the zero equality constraint in the relaxed OTLP: if $i \notin H^*$ and $\bar{i} \in (H^*)^n$, then $\bar{i}$ cannot lie in the incidence set $A_i$, so $S_{i,\bar{i}} = 0$ already.

Finally, combining our three new conditions with the feasibility constraints in the relaxed OTLP, any solution to the following system is an optimal solution $S^*$ in the relaxed OTLP, and vice versa:

$$
\begin{aligned}
& \sum_{\bar{i} \in \mathcal{V}^n} S_{i,\bar{i}}^* \leq p(i) \quad \forall i \in \mathcal{V}, && \sum_{i \in \mathcal{V}} S_{i,\bar{i}}^* \leq p_{\text{draft}}(\bar{i}) \quad \forall \bar{i} \in \mathcal{V}^n, \\
& \sum_{\bar{i} \in \mathcal{V}^n} S_{i,\bar{i}}^* = p(i) \quad \forall i \in H^*, && \sum_{i \in \mathcal{V}} S_{i,\bar{i}}^* = p_{\text{draft}}(\bar{i}) \quad \forall \bar{i} \notin (H^*)^n, \\
& S_{i,\bar{i}}^* = 0 \quad \forall i \in \mathcal{V}, \bar{i} \notin A_i, && S_{i,\bar{i}}^* = 0 \quad \forall i \in H^*, \bar{i} \notin (H^*)^n.
\end{aligned} \tag{42}
$$

Finally, note that some inequality constraints are redundant. We no longer require the $p(i)$ inequalities for $i \in H^*$, and we no longer require the $p_{\text{draft}}(\bar{i})$ inequalities for $\bar{i} \notin (H^*)^n$. This directly reduces the above system, which is equivalent to the relaxed OTLP, to the desired system. $\qquad\square$

## F   MAX-FLOW TRANSFORMATION DERIVATION

Although it is a standard network reduction technique to reduce row and column sum constraints to max-flow over a bipartite network, we provide a proof of Lemma F.1 here for completeness.

**Lemma F.1** (Bipartite-Flow Equivalence). *Let $G = (A \cup B, E \subseteq A \times B)$ be a bipartite graph, with $N(\cdot)$ denoting neighborhoods. Let $\boldsymbol{x} \in \mathbb{R}^A, \boldsymbol{y} \in \mathbb{R}^B, \boldsymbol{z} \in \mathbb{R}^E$ be nonnegative. Define a network $\Omega$ on nodes $\{s, t\} \cup A \cup B$, with capacity $x_a$ edges $s \to a$ for $a \in A$, capacity $y_b$ edges $b \to t$ for $b \in B$, and $\infty$-capacity edges $a \to b$ for $(a, b) \in E$. Then these two conditions are equivalent:*

*(1)* $\sum_{b \in N(a)} z_{(a,b)} = x_a$ *for all $a \in A$ and* $\sum_{a \in N(b)} z_{(a,b)} \leq y_b$ *for all $b \in B$,*

*(2)* $f(s \to a) = x_a$, $f(a \to b) = z_{(a,b)}$, $f(b \to t) = \sum_{a \in N(b)} z_{(a,b)}$ *is a maximal flow in $\Omega$.*

*Proof.* Suppose condition **(1)** holds. Then the flow assignment $f$ in the second condition satisfies

$$
f(s \to a) = x_a = \sum_{b \in N(a)} z_{(a,b)} = \sum_{b \in N(a)} f(a \to b), \tag{43}
$$

$$
f(b \to t) = \sum_{a \in N(b)} z_{(a,b)} = \sum_{a \in N(b)} f(a \to b). \tag{44}
$$

This means flow conservation holds. Also, $f(s \to a) \leq x_a$ for each $a \in A$, and $f(b \to t) \leq y_b$ for each $b \in B$ from the inequality constraints, so the capacity constraints hold. Thus, the assignment $f$ is feasible. In fact, its value is $\sum_{a \in A} f(s \to a) = \sum_{a \in A} x_a$, and no flow assignment in $\Omega$ can exceed this value since the total flow along edges $s \to a$ cannot exceed the sum of the capacities $x_a$. This shows the $f$ is indeed the maximal flow, so condition **(2)** is satisfied, as desired.

Conversely, suppose condition **(2)** holds. Then flow conservation at each $a \in A$ shows

$$
x_a = f(s \to a) = \sum_{b \in N(a)} f(a \to b) = \sum_{b \in N(a)} z_{(a,b)}. \tag{45}
$$

Also, the capacity at $b \to t$ shows $f(s \to b) = \sum_{a \in N(b)} z_{(a,b)} \leq y_b$. Thus, condition **(1)** holds. $\qquad\square$

## G  PROOFS OF RESIDUAL LPS

In this appendix, we prove the reduction the outer residual LP. We will use the following lemma. Note that each $x_a$ and $y_b$ must be nonnegative, as per the requirements of Lemma F.1.

**Lemma G.1.** *From Lemma F.1, consider condition (1) as a system in nonnegative variables $z_e$ with $x_a$-equality constraints and $y_b$-inequality constraints. This is feasible if and only if*

$$\sum_{a \in S} x_a \leq \sum_{b \in N(S)} y_b \quad \forall S \subseteq A. \tag{46}$$

*When $y_b$ constraints are equalities, feasibility is equivalent to the above and $\sum_{a \in A} x_a = \sum_{y \in B} y_b$.*

*Proof.* The first part follows from the equivalence of conditions (1) and (2) in Lemma F.1, and Gale's feasibility theorem for bipartite supply-demand networks applied to condition (2) (Gale, 1957). The second part holds as $\sum_{a \in A} x_a = \sum_{b \in B} y_b$ combines with condition (1),

$$\sum_{a \in A} x_a = \sum_{a \in A} \sum_{b \in N(a)} z_{(a,b)} = \sum_{(a,b) \in E} z_{(a,b)} = \sum_{b \in B} \sum_{a \in N(b)} z_{(a,b)} \leq \sum_{b \in B} y_b, \tag{47}$$

to force the equalities $\sum_{a \in N(b)} z_{(a,b)} = y_b$ for $b \in B$, as required. $\square$

Now, we prove the reduced LP for outer system residuals $p^{\text{res}}(i)$ in Theorem 6.1.

*Proof.* To solve for the residuals in the outer system, we need to find $0 \leq p_i \leq p(i)$ where

$$\sum_{\bar{i} \in \mathcal{V}^n \setminus (H^*)^n} S_{i,\bar{i}}^* = p_i \quad \forall i \in \mathcal{V} \setminus H^*, \qquad \sum_{i \in \mathcal{V} \setminus H^*} S_{i,\bar{i}}^* = p_{\text{draft}}(\bar{i}) \quad \forall \bar{i} \in \mathcal{V}^n \setminus (H^*)^n, \tag{48}$$

is feasible for nonnegative $S^*$. When this system is feasible, its solution $S^*$ will also be a solution to the outer system, so we can simply recover the residuals as $p^{\text{res}}(i) = p(i) - p_i$. Since all $p_i$ are nonnegative, we can use Lemma G.1, to show that the above system is feasible if and only if:

$$\sum_{i \in \mathcal{V} \setminus H^*} p_i = \sum_{\bar{i} \in \mathcal{V}^n \setminus (H^*)^n} p_{\text{draft}}(\bar{i}), \quad \sum_{i \in S} p_i \leq \sum_{\bar{i} \in N(S)} p_{\text{draft}}(\bar{i}) \quad \forall S \subseteq \mathcal{V} \setminus H^*, \tag{49}$$

where $N(S) \subseteq \mathcal{V}^n \setminus (H^*)^n$ denotes the neighborhood of $S \subseteq \mathcal{V} \setminus H^*$ in the bipartite graph with left vertices $\mathcal{V} \setminus H^*$, right vertices $\mathcal{V}^n \setminus (H^*)^n$, and edges $(i, \bar{i})$ with $i \in \text{set}(\bar{i})$. Note $N(S)$ contains precisely the $n$-tuples in $\mathcal{V}^n$ with at least one element in $S$. Therefore, we can write

$$\sum_{\bar{i} \in N(S)} p_{\text{draft}}(\bar{i}) - \sum_{i \in S} p(i) = 1 - \sum_{i \in S} p(i) - \sum_{\bar{i} \in (\mathcal{V} \setminus S)^n} p_{\text{draft}}(\bar{i}) = \psi(\mathcal{V} \setminus S). \tag{50}$$

Substituting this back and simplifying with $\psi$, our condition becomes:

$$\sum_{i \in \mathcal{V} \setminus H^*} p_i = \sum_{i \in \mathcal{V} \setminus H^*} p(i) + \psi(H^*), \quad \sum_{i \in S} p_i \leq \sum_{i \in S} p(i) + \psi(\mathcal{V} \setminus S) \quad \forall S \subseteq \mathcal{V} \setminus H^*. \tag{51}$$

Including the original constraints $0 \leq p_i \leq p(i)$ completes the proof. $\square$

## H  GREEDY POLYMATROID ALGORITHM FOR OUTER RESIDUAL LP

Here, we prove Theorem 6.2 by using polymatroids. Polymatroids are subsets of Euclidean spaces associated with submodular functions: they defined as the feasibility set of subset sum inequality constraints with bounds being evaluations of the submodular function. Thus, they are directly related to the outer residual LP. In our proof, we refer only to standard polymatroid theory from Chapter 44 in Schrijver et al. (2003).

*Proof.* Because $\psi(S)$ is submodular in $S$, and $\sum_{i \in S} p(i)$ is modular (additive) in $S$, then

$$\phi(S) = \sum_{i \in S} p(i) + \psi(\mathcal{V} \setminus S) = 1 - \sum_{\bar{i} \in (\mathcal{V} \setminus S)^n} p_{\text{draft}}(\bar{i}) \tag{52}$$

is submodular in $S$. Therefore, solving Theorem 6.1 amounts to finding $\boldsymbol{x} \in EP_\phi \cap B_p$ with $\boldsymbol{x} \succeq \boldsymbol{0}$ and $\boldsymbol{1} \cdot \boldsymbol{x} = \phi(\mathcal{V} \setminus H^*)$, where $B_p$ is the feasibility set of all constraints $p_i \leq p(i)$, and $EP_\phi$ is the *extended polymatroid* associated with the submodular function $\phi$:

$$EP_\phi = \left\{ \boldsymbol{x} : \sum_{i \in S} x_i \leq \phi(S) \quad \forall S \subseteq \mathcal{V} \setminus H^* \right\}. \tag{53}$$

By Equations 44.8 and 44.9 in Schrijver et al. (2003), the box-polymatroid intersection $EP_\phi \cap B_p$ is also an extended polymatroid $EP_{\phi|p}$, associated with the following submodular function $\phi|p$:

$$(\phi|p)(S) = \min_{T \subseteq S} \left\{ \sum_{i \in S \setminus T} p(i) + \phi(T) \right\} = \sum_{i \in S} p(i) + \min_{\mathcal{V} \setminus S \subseteq T} \psi(T). \tag{54}$$

Since our system is feasible, and has the inequality constraint $\boldsymbol{1} \cdot \boldsymbol{x} \leq \phi(\mathcal{V} \setminus H^*)$ and equality constraint $\boldsymbol{1} \cdot \boldsymbol{x} = \phi(\mathcal{V} \setminus H^*)$, we can replace the equality constraint with an objective $\max \boldsymbol{1} \cdot \boldsymbol{x}$. This turns Theorem 6.1 into the following extended polymatroid optimization problem:

$$\max \boldsymbol{1} \cdot \boldsymbol{x} \quad \text{s.t} \quad \boldsymbol{x} \in EP_{\phi|p}, \quad \boldsymbol{x} \succeq 0. \tag{55}$$

We first ignore the nonnegativity constraint. To solve this simplified version, we can use the two-step greedy algorithm from Theorem 44.3 in Schrijver et al. (2003). First, we reorder $\mathcal{V} \setminus H^*$ in increasing order of the weights $\boldsymbol{1}$. This can be arbitrary, so we specify it as the given ordering $\{v_1, \ldots, v_k\}$. Then, we compute consecutive differences $x_{v_i} = (\phi|p)(S_i) - (\phi|p)(S_{i-1})$, where $S_i = \{v_1, \ldots, v_i\}$ is the $i$th prefix of $\mathcal{V} \setminus H^*$. We can compute this explicitly as

$$x_{v_i} = \sum_{i \in S_i} p(i) + \min_{\mathcal{V} \setminus S_i \subseteq T} \psi(T) - \sum_{i \in S_{i-1}} p(i) - \min_{\mathcal{V} \setminus S_{i-1} \subseteq T} \psi(T) \tag{56}$$

$$= p(v_i) + \min_{H^* \cup \{v_{i+1}, \ldots, v_k\} \subseteq T} \psi(T) - \min_{H^* \cup \{v_i, \ldots, v_k\} \subseteq T} \psi(T) \tag{57}$$

$$= p(v_i) + \min_{H_{i+1} \subseteq T} \psi(T) - \min_{H_i \subseteq T} \psi(T). \tag{58}$$

This matches the desired expression, so to complete the proof, it suffices to show $\boldsymbol{x} \succeq 0$, i.e. each $x_{v_i} \geq 0$. Indeed, suppose $T^* \supseteq H_{i+1}$ minimizes the first of the two min $\psi$ terms above. Since $H_i = H_{i+1} \cup \{v_i\}$, we will have $T^* \cup \{v_i\} \supseteq H_i$, and thus

$$\min_{H_i \subseteq T} \psi(T) \leq \psi(T^* \cup \{v_i\}) \tag{59}$$

$$= \sum_{x \in (T^* \cup \{v_i\})} p(x) - \sum_{\bar{i} \in (T^* \cup \{v_i\})^n} p_{\text{draft}}(\bar{i}) \tag{60}$$

$$\leq p(v_i) + \sum_{x \in T^*} p(x) - \sum_{\bar{i} \in (T^*)^n} p_{\text{draft}}(\bar{i}) \tag{61}$$

$$= p(v_i) + \psi(T^*) = p(v_i) + \min_{H_{i+1} \subseteq T} \psi(T). \tag{62}$$

Rearranging this exactly gives $x_{v_i} \geq 0$, as required. $\square$

## I  GLOBAL OUTER SOLUTION

Here, we prove Theorem 6.4. We will use the following lemma, which relates the minimization of feasibility of a transport system to the nonnegativity of an associated convex function.

**Lemma I.1** (Convex Transport Solver 1). *Let $G = (A \cup B, E \subseteq A \times B)$ be a bipartite graph without isolated points (each neighborhood is nonempty). Suppose the following system is feasible in nonnegative variables $S_{a,b}$ for $(a,b) \in E$:*

$$\sum_{b \in N(a)} S_{a,b} = x_a \quad \forall a \in A, \qquad \sum_{a \in N(b)} S_{a,b} = y_b \quad \forall b \in B. \tag{63}$$

*Then the following function $\Phi : \mathbb{R}^{|A|} \to \mathbb{R}$ is convex and nonnegative:*

$$\Phi((\alpha_a)_{a \in A}) = \sum_{b \in B} y_b \log\left(\sum_{a \in N(b)} e^{\alpha_a}\right) - \sum_{a \in A} x_a \alpha_a. \tag{64}$$

*Furthermore, for any $\epsilon > 0$, there exists $(\alpha_a)_{a \in A}$ with $\|\nabla \Phi((\alpha_a)_{a \in A})\|_1 \leq \epsilon$.*

*Proof.* We will use the Hall-type conditions from Lemma G.1. By feasibility of the system,

$$\sum_{b \in N(S)} y_b \geq \sum_{a \in S} x_a \quad \forall S \subseteq A, \qquad \sum_{b \in B} y_b = \sum_{a \in A} x_a. \tag{65}$$

Now, it is clear that $\Phi$ is convex in $\alpha$ since it is a linear combination of terms convex in $\alpha$ (LogSumExp and linear). To show it is nonnegative at a particular point $\alpha$, relabel $A = \{1, 2, \ldots, m\}$ such that $\alpha_1 \geq \ldots \geq \alpha_m$. For each $b \in B$, we have the lower bound

$$\log\left(\sum_{a \in N(b)} e^{\alpha_a}\right) \geq \max_{a \in N(b)} \alpha_a. \tag{66}$$

Now, let $N_k = N(\{1, \ldots, k\}) \setminus N(\{1, \ldots, k-1\})$ for each $k \in A$. We have $\max_{a \in N(b)} \alpha_a = \alpha_k$ for each $b \in N_k$, because $k \in N(b)$ and $1, \ldots, k-1 \notin N(b)$. Furthermore, the sets $N_1, \ldots, N_m$ are disjoint, with union $N(\{1, \ldots, m\}) = B$. Using this, we can lower bound

$$\Phi((\alpha_a)_{a \in A}) \geq \sum_{b \in B} y_b \max_{a \in N(b)} \alpha_a - \sum_{a \in A} x_a \alpha_a \tag{67}$$

$$= \sum_{k=1}^{m} \sum_{b \in N_k} y_b \alpha_k - \sum_{a=1}^{m} x_a \alpha_a \tag{68}$$

$$= \sum_{a=1}^{m} \left(\sum_{b \in N_a} y_b - x_a\right) \alpha_a. \tag{69}$$

Denote the coefficient of $\alpha_a$ by $c_a$. For any $1 \leq k \leq m$, we see by applying the Hall conditions that

$$\sum_{a=1}^{k} c_a = \sum_{b \in N_1 \cup \ldots \cup N_k} y_b - \sum_{a=1}^{k} x_a \tag{70}$$

$$= \sum_{b \in N(\{1, \ldots, k\})} y_b - \sum_{a \in \{1, \ldots, k\}} x_a \geq 0, \tag{71}$$

with equality at $k = m$. Thus, as each $\alpha_j \geq \alpha_{j+1}$, we have the desired nonnegativity:

$$\Phi((\alpha_a)_{a \in A}) \geq \sum_{a=1}^{m} c_a \alpha_a = \sum_{a=1}^{m} c_a (\alpha_a - \alpha_m) \tag{72}$$

$$= \sum_{a=1}^{m} \sum_{j=a+1}^{m} c_a (\alpha_{j-1} - \alpha_j) \tag{73}$$

$$= \sum_{j=1}^{m-1} \sum_{a=1}^{j} c_a (\alpha_j - \alpha_{j+1}) \tag{74}$$

$$= \sum_{j=1}^{m-1} \left(\sum_{a=1}^{j} c_a\right) (\alpha_j - \alpha_{j+1}) \geq 0. \tag{75}$$

Because $\Phi$ is convex and bounded below, by the Brøndsted–Rockafellar approximation theorem (Brøndsted & Rockafellar, 1965), there are points on $\Phi$ with arbitrarily small subgradients. As $\Phi$ is continuously differentiable, then for each $\epsilon > 0$, some $(\alpha_a)_{a \in A}$ satisfies $\|\nabla\Phi(\alpha)\|_1 \le \epsilon$. $\qquad\square$

For the outer system, we can apply Lemma I.1 with bipartite components $A = \mathcal{V} \setminus H^*$ and $B = \mathcal{V}^n \setminus (H^*)^n$, edges $(a, b) \in E$ if and only if $a \in \text{set}(b)$, and bounds $x_a = p_a$ for all $a \in A$ and $y_b = p_{\text{draft}}(b)$ for all $b \in B$. (Here, $p_a$ are the outer residual LP variables we solved for in Theorem 6.1, given by $p_a = p(a) - p^{\text{res}}(a)$.) The key idea is that the gradient norm bound from the lemma bounds the total deviation from $p(i)$ equality constraints in the outer system, which allows us to prove Theorem 6.4.

*Proof.* We first show that $\|\nabla\Phi_T\|_1 \le \epsilon + \epsilon_T$ is feasible. Define $\Phi : \mathbb{R}^{\mathcal{V} \setminus H^*} \to \mathbb{R}$ as in Lemma I.1:

$$\Phi((\alpha_i)_{i \in \mathcal{V} \setminus H^*}) = \sum_{\bar{i} \in \mathcal{V}^n \setminus (H^*)^n} p_{\text{draft}}(\bar{i}) \log \left( \sum_{i \in \text{set}(\bar{i}) \setminus H^*} e^{\alpha_i} \right) - \sum_{i \in \mathcal{V} \setminus H^*} p_i \alpha_i. \tag{76}$$

By the lemma, there is some $(\alpha_i^*)_{i \in \mathcal{V} \setminus H^*}$ with $\|\nabla\Phi((\alpha_i^*)_{i \in \mathcal{V} \setminus H^*})\|_1 \le \epsilon$. Thus,

$$\sum_{i \in T} \left| \frac{\partial \Phi(\alpha_i^*)}{\partial \alpha_i} \right| \le \sum_{i \in \mathcal{V} \setminus H^*} \left| \frac{\partial \Phi(\alpha_i^*)}{\partial \alpha_i} \right| \le \epsilon. \tag{77}$$

We can also compute the absolute difference in the $i$th partial derivatives of $\Phi, \Phi_T$ for any $i \in T$ as

$$\left| \frac{\partial \Phi_T(\alpha_i^*)}{\partial \alpha_i} - \frac{\partial \Phi(\alpha_i^*)}{\partial \alpha_i} \right| = \sum_{\bar{i} \in V^n \setminus (H^* \cup T)^n} \frac{e^{\alpha_i^*} \cdot \mathbf{1}[i \in \text{set}(\bar{i}) \setminus H^*]}{\sum_{j \in \text{set}(\bar{i}) \setminus H^*} e^{\alpha_j^*}} \cdot p_{\text{draft}}(\bar{i}). \tag{78}$$

Thus, summing the above over all $i \in T$ gives

$$\sum_{i \in T} \left| \frac{\partial \Phi_T(\alpha_i^*)}{\partial \alpha_i} - \frac{\partial \Phi(\alpha_i^*)}{\partial \alpha_i} \right| = \sum_{\bar{i} \in \mathcal{V}^n \setminus (H^* \cup T)^n} \sum_{i \in T} \frac{e^{\alpha_i^*} \cdot \mathbf{1}[i \in \text{set}(\bar{i}) \setminus H^*]}{\sum_{j \in \text{set}(\bar{i}) \setminus H^*} e^{\alpha_j^*}} \cdot p_{\text{draft}}(\bar{i}) \tag{79}$$

$$\le \sum_{\bar{i} \in \mathcal{V}^n \setminus (H^* \cup T)^n} p_{\text{draft}}(\bar{i}) = 1 - \left( \sum_{i \in H^* \cup T} q(i) \right)^n = \epsilon_T. \tag{80}$$

Because $\Phi_T$ is constant over $\alpha_i$ for $i \notin T$, and thus has zero partial derivative at these variables, the Triangle Inequality implies that $\|\nabla\Phi_T\|_1$ falls below $\epsilon + \epsilon_T$ at $(\alpha_i^*)_{i \in \mathcal{V} \setminus H^*}$:

$$\|\nabla\Phi_T((\alpha_i^*)_{i \in \mathcal{V} \setminus H^*})\|_1 \le \sum_{i \in T} \left| \frac{\partial \Phi(\alpha_i^*)}{\partial \alpha_i} \right| + \sum_{i \in T} \left| \frac{\partial \Phi_T(\alpha_i^*)}{\partial \alpha_i} - \frac{\partial \Phi(\alpha_i^*)}{\partial \alpha_i} \right| \le \epsilon + \epsilon_T. \tag{81}$$

Now, we prove the second part of the theorem for arbitrary $(\alpha_i)_{i \in \mathcal{V} \setminus H^*}$. First, we bound the sum of the partial derivatives of $\Phi$ over $i \in \mathcal{V} \setminus (H^* \cup T)$:

$$\sum_{i \in \mathcal{V} \setminus (H^* \cup T)} \left| \frac{\partial \Phi}{\partial \alpha_i} \right| = \sum_{i \in \mathcal{V} \setminus (H^* \cup T)} \left| \sum_{\substack{\bar{i} \in \mathcal{V}^n \setminus (H^*)^n \\ i \in \text{set}(\bar{i}) \setminus H^*}} \frac{e^{\alpha_i}}{\sum_{j \in \text{set}(\bar{i}) \setminus H^*} e^{\alpha_j}} \cdot p_{\text{draft}}(\bar{i}) - p_i \right| \tag{82}$$

$$\le \sum_{i \in \mathcal{V} \setminus (H^* \cup T)} \left( \sum_{\substack{\bar{i} \in \mathcal{V}^n \setminus (H^* \cup T)^n \\ i \in \text{set}(\bar{i}) \setminus H^*}} \frac{e^{\alpha_i}}{\sum_{j \in \text{set}(\bar{i}) \setminus H^*} e^{\alpha_j}} \cdot p_{\text{draft}}(\bar{i}) + p_i \right) \tag{83}$$

$$\le \sum_{\bar{i} \in \mathcal{V}^n \setminus (H^* \cup T)^n} p_{\text{draft}}(\bar{i}) + \sum_{i \in V \setminus (H^* \cup T)} p_i \tag{84}$$

$$\le 2 \sum_{\bar{i} \in \mathcal{V}^n \setminus (H^* \cup T)^n} p_{\text{draft}}(\bar{i}) = 2\epsilon_T, \tag{85}$$

where the last inequality holds by Equation (49). Therefore, we have by the Triangle Inequality and Equation (80) (holds for any point) that

$$\|\nabla\Phi((\alpha_i)_{i\in\mathcal{V}\setminus H^*})\|_1 = \sum_{i\in T}\left|\frac{\partial\Phi}{\partial\alpha_i}\right| + \sum_{i\in(\mathcal{V}\setminus H^*)\setminus T}\left|\frac{\partial\Phi}{\partial\alpha_i}\right| \tag{86}$$

$$\leq \|\nabla\Phi_T((\alpha_i)_{i\in\mathcal{V}\setminus H^*})\|_1 + \sum_{i\in T}\left|\frac{\partial\Phi_T(\alpha_i^*)}{\partial\alpha_i} - \frac{\partial\Phi(\alpha_i^*)}{\partial\alpha_i}\right| + 2\epsilon_T \tag{87}$$

$$\leq \|\nabla\Phi_T((\alpha_i)_{i\in\mathcal{V}\setminus H^*})\|_1 + 3\epsilon_T. \tag{88}$$

Finally, using the explicit representation for partial derivatives of $\Phi$, and substituting in $S_{i,\bar{i}}$ from Equation (14), we get

$$\sum_{i\in\mathcal{V}\setminus H^*}\left|\sum_{\substack{\bar{i}\in\mathcal{V}^n\setminus(H^*)^n\\i\in\text{set}(\bar{i})\setminus H^*}}\frac{e^{\alpha_i}}{\sum_{j\in\text{set}(\bar{i})\setminus H^*}e^{\alpha_j}}\cdot p_{\text{draft}}(\bar{i}) - p_i\right| = \sum_{i\in\mathcal{V}\setminus H^*}\left|\sum_{\substack{\bar{i}\in\mathcal{V}^n\setminus(H^*)^n\\i\in\text{set}(\bar{i})\setminus H^*}}S_{i,\bar{i}} - p_i\right| \tag{89}$$

$$\leq \|\nabla\Phi_T((\alpha_i)_{i\in\mathcal{V}\setminus H^*})\|_1 + 3\epsilon_T, \tag{90}$$

hence $S_{i,\bar{i}}$ satisfy the $p_i$ equality constraints of the outer system with up to $\|\nabla\Phi_T((\alpha_i)_{i\in\mathcal{V}\setminus H^*})\|_1 + 3\epsilon_T$ total leeway. They also satisfy the $p_{\text{draft}}(\bar{i})$ equality constraints exactly, as

$$\sum_{i\in\text{set}(\bar{i})\setminus H^*}S_{i,\bar{i}} = \sum_{i\in\text{set}(\bar{i})\setminus H^*}\frac{e^{\alpha_i}}{\sum_{j\in\text{set}(\bar{i})\setminus H^*}e^{\alpha_j}}\cdot p_{\text{draft}}(\bar{i}) = p_{\text{draft}}(\bar{i}). \tag{91}$$

This completes the proof. $\qquad\square$

## J  GLOBAL INNER SOLUTION

Now, we prove Theorem 6.5. We will use the lemma below, which is quite similar to Lemma I.1 in Appendix I, except that it contains an extra slack term in the LogSumExp to deal with a bipartite transport system with equalities on one side and inequalities on the other side.

**Lemma J.1** (Convex Transport Solver 2). *Let $G = (A\cup B, E\subseteq A\times B)$ be a bipartite graph without isolated points (each neighborhood is nonempty). Suppose the following system is feasible in nonnegative variables $S_{a,b}$ for $(a,b)\in E$:*

$$\sum_{b\in N(a)}S_{a,b} = x_a \quad\forall a\in A, \qquad \sum_{a\in N(b)}S_{a,b} \leq y_b \quad\forall b\in B. \tag{92}$$

*Then the following function $\Theta:\mathbb{R}^{|A|}\to\mathbb{R}$ is convex and nonnegative:*

$$\Theta((\alpha_a)_{a\in A}) = \sum_{b\in B}y_b\log\left(1 + \sum_{a\in N(b)}e^{\alpha_a}\right) - \sum_{a\in A}x_a\alpha_a. \tag{93}$$

*Furthermore, for any $\epsilon > 0$, there exists $(\alpha_a)_{a\in A}$ with $\|\nabla\Theta((\alpha_a)_{a\in A})\|_1 \leq \epsilon$.*

*Proof.* We add a vertex $a_0$ to $A\subseteq G$, with edges from $a_0$ to all $b\in B$, to form the graph $G' = (A\cup\{a_0\}\cup B, E')$. Now, let $S_{a,b}$ for $(a,b)\in E$ satisfy the given system. We define $S'_{a,b}$ for $(a,b)\in E'$, such that $S'_{a,b} = S_{a,b}$ for $a\neq a_0$ and $S'_{a_0,b} = y_b - \sum_{a\in N(b)}S_{a,b} \geq 0$. Also, define $x'_a$ for all $a\in A\cup\{a_0\}$ such that $x'_a = x_a$ if $a\neq a_0$ and $x'_{a_0} = \sum_{b\in B}y_b - \sum_{a\in A}x_a \geq 0$, and define $y'_b$ for all $b\in B$ such that $y'_b = y_b$. Then we see the following, where $N'(\cdot)$ now denotes $G'$-neighborhoods:

$$\sum_{b\in N'(a_0)}S'_{a_0,b} = \sum_{b\in B}\left(y_b - \sum_{a\in N(b)}S_{a,b}\right) = \sum_{b\in B}y_b - \sum_{a\in A}\sum_{b\in N(a)}S_{a,b} = x'_{a_0} \tag{94}$$

and for each $b \in B$, we have

$$\sum_{a \in N'(b)} S'_{a,b} = \sum_{a \in N(b), a \neq a_0} S_{a,b} + S_{a_0,b} = y_b = y'_b. \tag{95}$$

This shows the following system is feasible:

$$\sum_{b \in N'(a)} S'_{a,b} = x'_a \quad \forall a \in A \cup \{a_0\}, \qquad \sum_{a \in N'(b)} S'_{a,b} = y'_b \quad \forall b \in B. \tag{96}$$

Hence, we can apply Lemma I.1 to show that

$$\Phi((\alpha_a)_{a \in A \cup \{a_0\}}) = \sum_{b \in B} y'_b \log \left( \sum_{a \in N'(b)} e^{\alpha_a} \right) - \sum_{a \in A \cup \{a_0\}} x'_a \alpha_a \tag{97}$$

is convex and nonnegative. Restricting this to the slice $\alpha_{a_0} = 0$ gives $\Theta((\alpha_a)_{a \in A})$ as defined in the theorem, as each $N'(b)$ contains $a_0$, corresponding to the $+1$ term in log, and the $x'_{a_0} \alpha_{a_0}$ contribution cancels. Thus, $\Theta$ is also convex and nonnegative. The existence of arbitrarily small gradient norms then follows from the same argument as in the proof of Lemma I.1. $\qquad\square$

For the inner system, we can apply Lemma J.1 with bipartite components $A = H^*$ and $B = (H^*)^n$, edges $(a, b) \in E$ if and only if $a \in \text{set}(b)$, and bounds $x_a = p(a)$ for all $a \in A$ and $y_b = p_{\text{draft}}(b)$ for all $b \in B$. Once more, the gradient norm bound from the lemma bounds the total deviation from $p(i)$ equality constraints in the inner system, proving Theorem 6.5.

*Proof.* We first show that $\|\nabla \Theta_T\|_1 \leq \epsilon + \gamma_T$ is feasible. Define $\Theta : \mathbb{R}^{H^*} \to \mathbb{R}$ as in Lemma J.1:

$$\Theta((\alpha_i)_{i \in H^*}) = \sum_{\bar{i} \in (H^*)^n} p_{\text{draft}}(\bar{i}) \log \left( 1 + \sum_{i \in \text{set}(\bar{i})} e^{\alpha_i} \right) - \sum_{i \in H^*} p(i)\alpha_i. \tag{98}$$

By the lemma, there is some $(\alpha_i^*)_{i \in H^*}$ with $\|\nabla \Theta((\alpha_i^*)_{i \in H^*})\|_1 \leq \epsilon$. Thus,

$$\sum_{i \in T} \left| \frac{\partial \Theta(\alpha_i^*)}{\partial \alpha_i} \right| \leq \sum_{i \in H^*} \left| \frac{\partial \Theta(\alpha_i^*)}{\partial \alpha_i} \right| \leq \epsilon. \tag{99}$$

We can also compute the absolute difference in the $i$th partial derivatives of $\Theta, \Theta_T$ for any $i \in T$ as

$$\left| \frac{\partial \Theta_T(\alpha_i^*)}{\partial \alpha_i} - \frac{\partial \Theta(\alpha_i^*)}{\partial \alpha_i} \right| = \sum_{\bar{i} \in (H^*)^n \backslash T^n} \frac{e^{\alpha_i^*} \cdot \mathbf{1}[i \in \text{set}(\bar{i})]}{1 + \sum_{j \in \text{set}(\bar{i})} e^{\alpha_j^*}} \cdot p_{\text{draft}}(\bar{i}). \tag{100}$$

Thus, summing the above over all $i \in T$ gives

$$\sum_{i \in T} \left| \frac{\partial \Theta_T(\alpha_i^*)}{\partial \alpha_i} - \frac{\partial \Theta(\alpha_i^*)}{\partial \alpha_i} \right| = \sum_{\bar{i} \in (H^*)^n \backslash T^n} \sum_{i \in T} \frac{e^{\alpha_i^*} \cdot \mathbf{1}[i \in \text{set}(\bar{i})]}{1 + \sum_{j \in \text{set}(\bar{i})} e^{\alpha_j^*}} \cdot p_{\text{draft}}(\bar{i}). \tag{101}$$

$$\leq \sum_{\bar{i} \in (H^*)^n \backslash T^n} p_{\text{draft}}(\bar{i}) = \left( \sum_{i \in H^*} q(i) \right)^n - \left( \sum_{i \in T} q(i) \right)^n = \gamma_T. \tag{102}$$

Because $\Theta_T$ is constant over $\alpha_i$ for $i \notin T$, and thus has zero partial derivative at these variables, the Triangle Inequality implies that $\|\nabla \Theta_T\|_1$ falls below $\epsilon + \gamma_T$ at $(\alpha_i^*)_{i \in H^*}$:

$$\|\nabla \Theta_T((\alpha_i^*)_{i \in H^*})\|_1 \leq \sum_{i \in T} \left| \frac{\partial \Theta(\alpha_i^*)}{\partial \alpha_i} \right| + \sum_{i \in T} \left| \frac{\partial \Theta_T(\alpha_i^*)}{\partial \alpha_i} - \frac{\partial \Theta(\alpha_i^*)}{\partial \alpha_i} \right| \leq \epsilon + \gamma_T. \tag{103}$$

Now, we prove the second part of the theorem for arbitrary $(\alpha_i)_{i \in H^*}$. First, we bound the sum of the partial derivatives of $\Theta$ over $i \in H^* \setminus T$:

$$\sum_{i \in H^* \setminus T} \left| \frac{\partial \Theta}{\partial \alpha_i} \right| = \sum_{i \in H^* \setminus T} \left| \sum_{\substack{\bar{i} \in (H^*)^n \\ i \in \text{set}(\bar{i})}} \frac{e^{\alpha_i}}{1 + \sum_{j \in \text{set}(\bar{i})} e^{\alpha_j}} \cdot p_{\text{draft}}(\bar{i}) - p(i) \right| \tag{104}$$

$$\leq \sum_{i \in H^* \setminus T} \left( \sum_{\substack{\bar{i} \in (H^*)^n \\ i \in \text{set}(\bar{i})}} \frac{e^{\alpha_i}}{1 + \sum_{j \in \text{set}(\bar{i})} e^{\alpha_j}} \cdot p_{\text{draft}}(\bar{i}) + p(i) \right) \tag{105}$$

$$\leq \sum_{\bar{i} \in (H^*)^n \setminus T^n} p_{\text{draft}}(\bar{i}) + \sum_{i \in H^* \setminus T} p(i) \tag{106}$$

$$\leq 2 \sum_{\bar{i} \in (H^*)^n \setminus T^n} p_{\text{draft}}(\bar{i}) = 2\gamma_T, \tag{107}$$

where the last inequality using the fact that $\psi$ is minimized at $H^*$:

$$0 \leq \psi(T) - \psi(H^*) \tag{108}$$

$$= \sum_{i \in T} p(i) - \sum_{\bar{i} \in T^n} p_{\text{draft}}(\bar{i}) - \sum_{i \in H^*} p(i) + \sum_{\bar{i} \in (H^*)^n} p_{\text{draft}}(\bar{i}) \tag{109}$$

$$= \sum_{\bar{i} \in (H^*)^n \setminus T^n} p_{\text{draft}}(\bar{i}) - \sum_{i \in H^* \setminus T} p(i) = \gamma_T - \sum_{i \in H^* \setminus T} p(i). \tag{110}$$

Therefore, we have by the Triangle Inequality and Equation (102) (holds for any point) that

$$\|\nabla \Theta((\alpha_i)_{i \in H^*})\|_1 = \sum_{i \in T} \left| \frac{\partial \Theta}{\partial \alpha_i} \right| + \sum_{i \in H^* \setminus T} \left| \frac{\partial \Theta}{\partial \alpha_i} \right| \tag{111}$$

$$\leq \|\nabla \Theta_T((\alpha_i)_{i \in H^*})\|_1 + \sum_{i \in T} \left| \frac{\partial \Theta_T(\alpha_i^*)}{\partial \alpha_i} - \frac{\partial \Theta(\alpha_i^*)}{\partial \alpha_i} \right| + 2\gamma_T \tag{112}$$

$$\leq \|\nabla \Theta_T((\alpha_i)_{i \in H^*})\|_1 + 3\gamma_T. \tag{113}$$

Finally, using the explicit representation for partial derivatives of $\Theta$, and substituting in $S_{i,\bar{i}}$ from Equation (17), we get

$$\sum_{i \in H^*} \left| \sum_{\substack{\bar{i} \in (H^*)^n \\ i \in \text{set}(\bar{i})}} \frac{e^{\alpha_i}}{1 + \sum_{j \in \text{set}(\bar{i})} e^{\alpha_j}} \cdot p_{\text{draft}}(\bar{i}) - p(i) \right| = \sum_{i \in H^*} \left| \sum_{\substack{\bar{i} \in (H^*)^n \\ i \in \text{set}(\bar{i})}} S_{i,\bar{i}} - p(i) \right| \tag{114}$$

$$\leq \|\nabla \Theta_T((\alpha_i)_{i \in H^*})\|_1 + 3\gamma_T, \tag{115}$$

hence $S_{i,\bar{i}}$ satisfy the $p_i$ equality constraints of the inner system with up to $\|\nabla \Theta_T((\alpha_i)_{i \in H^*})\|_1 + 3\gamma_T$ total leeway. They also satisfy the $p_{\text{draft}}(\bar{i})$ inequality constraints, as

$$\sum_{i \in \text{set}(\bar{i})} S_{i,\bar{i}} = \sum_{i \in \text{set}(\bar{i})} \frac{e^{\alpha_i}}{1 + \sum_{j \in \text{set}(\bar{i})} e^{\alpha_j}} \cdot p_{\text{draft}}(\bar{i}) \leq p_{\text{draft}}(\bar{i}). \tag{116}$$

This completes the proof. $\qquad \square$

## K  TRUNCATED SOLVER DETAILS

In this section, we discuss the specific implementation details for convex minimization in our inner and outer solvers, including under what conditions our solver fails.

The first step is to group coefficients of the same LogSumExp terms, to obtain a more compact representation of $\Phi_T$ and $\Theta_T$. For $\Phi_T$ in the outer system, the coefficient of a term containing precisely $\alpha_{i_1}, \ldots, \alpha_{i_k}$ (none of $i_1, \ldots, i_k$ lie in $H^*$) is explicitly

$$\sum_{\substack{\bar{i} \in (H^* \cup T)^n \setminus (H^*)^n \\ \text{set}(\bar{i}) \setminus H^* = \{i_1, \ldots, i_k\}}} p_{\text{draft}}(\bar{i}) = \sum_{A \subseteq \{i_1, \ldots, i_k\}} (-1)^{k-|A|} \sum_{i \in (H^* \cup A)^n} p_{\text{draft}}(\bar{i}) \tag{117}$$

$$= \sum_{A \subseteq \{i_1, \ldots, i_k\}} (-1)^{k-|A|} \left( \sum_{i \in H^* \cup A} q(i) \right)^n . \tag{118}$$

The second expression is a consequence of the principle of inclusion and exclusion (PIE). The last expression can be computed efficiently using dynamic programming. Similarly, for $\Theta_T$ in the inner system, the coefficient of a term containing precisely $\alpha_{i_1}, \ldots, \alpha_{i_k}$ is

$$\sum_{\substack{\bar{i} \in T^n \\ \text{set}(\bar{i}) = \{i_1, \ldots, i_k\}}} p_{\text{draft}}(\bar{i}) = \sum_{A \subseteq \{i_1, \ldots, i_k\}} (-1)^{k-|A|} \sum_{i \in A^n} p_{\text{draft}}(\bar{i}) \tag{119}$$

$$= \sum_{A \subseteq \{i_1, \ldots, i_k\}} (-1)^{k-|A|} \left( \sum_{i \in A} q(i) \right)^n . \tag{120}$$

Again, this formula can be derived with PIE, and can be computed efficiently with dynamic programming. Once we have grouped coefficients, both $\Phi_T, \Theta_T$ have at most

$$\sum_{i=1}^{n} \binom{|T|}{i} \sim \frac{|T|^n}{n!} \tag{121}$$

terms. With this compact representation, one can quickly compute the gradients of $\Phi_T$ and $\Theta_T$. We manually implement a function that returns the value of each of these functions and the corresponding gradient at an input point. Then, we use the the standard L-BFGS-B minimizer from SciPy (Virtanen et al., 2020) to run gradient descent, to converge to a point with near-zero gradient norm. This is an ideal algorithm for gradient descent because it converges quite fast for convex functions in few variables, and we require execution on the order of milliseconds. The SciPy API also returns the final gradient norm value, and allows early termination when the gradient norm falls below a threshold. To ensure our approach remains on the order of milliseconds, we limit L-BFGS-B to 25 iterations.

Finally, we discuss the issue of early termination, i.e. solve failure. There are two situations to consider: **(a)** $T$ is too large, **(b)** the gradient norm does not fall below the desired threshold. For **(a)**, we have set the following hard limits on $|T|$ through experimentation: 50 for $n = 2$, 20 for $n = 3$, 10 for $n = 4$, and 10 for $n = 5$. Note that these numbers are quite conservative: future work could aim to improve runtime and reduce early terminations by using a non-fixed threshold, or an alternative gradient descent method that works better for larger $T$. For **(b)**, because the SciPy API terminates when the threshold is reached, this can only occur if the maximum iterations are reached, and the gradient norm is too large. In practice, we find this is rarely the case: $|T|$ being too large is usually the reason for failure.

## L    APPROXIMATION GUARANTEES

Here, we prove Lemma 6.6, which translates the inner and outer solver deviations into concrete bounds on deviations in the approximate OTLP solution.

*Proof.* Define the row-wise sums $p'_i = \sum_{i \in \text{set}(\bar{i})} S_{i,\bar{i}}$. Also, denote the residuals $r(\bar{i}) = p_{\text{draft}}(\bar{i}) - \sum_{i \in \text{set}(\bar{i})} S_{i,\bar{i}}$. From Theorem 6.5 and Theorem 6.4, we know that $\sum_{i \in H^*} |p'_i - p(i)| \leq \beta$ and $\sum_{i \in \mathcal{V} \setminus H^*} |p'_i - p_i| \leq \alpha$. We first show that the $p_{\text{draft}}(\bar{i})$ equalities in the OTLP hold. For $\bar{i} \in (H^*)^n$,

we have

$$\sum_{i \in \mathcal{V}} C_{i,\bar{\imath}} = \sum_{i \in \text{set}(\bar{\imath})} S_{i,\bar{\imath}} + \sum_{i \in \mathcal{V} \setminus H^*} \frac{p(i) - p_i}{\sum_{j \in \mathcal{V} \setminus H^*}(p(j) - p_j)} \cdot r(\bar{\imath}) \tag{122}$$

$$= \sum_{i \in \text{set}(\bar{\imath})} S_{i,\bar{\imath}} + r(\bar{\imath}) = p_{\text{draft}}(\bar{\imath}). \tag{123}$$

Also, for $\bar{\imath} \notin (H^*)^n$, using the equality guarantee of the outer solver,

$$\sum_{i \in \mathcal{V}} C_{i,\bar{\imath}} = \sum_{i \in \text{set}(\bar{\imath})} S_{i,\bar{\imath}} = p_{\text{draft}}(\bar{\imath}). \tag{124}$$

Now, we bound the deviation from the $p(i)$ equality constraints in the OTLP. We have for all $i \in H^*$ that

$$\sum_{\bar{\imath} \in \mathcal{V}^n} C_{i,\bar{\imath}} = \sum_{\bar{\imath} \in (H^*)^n, i \in \text{set}(\bar{\imath})} S_{i,\bar{\imath}} = p'_i, \tag{125}$$

and we have for all $i \notin H^*$ that

$$\sum_{\bar{\imath} \in \mathcal{V}^n} C_{i,\bar{\imath}} = \sum_{\bar{\imath} \in \mathcal{V}^n \setminus (H^*)^n, i \in \text{set}(\bar{\imath}) \setminus H^*} S_{i,\bar{\imath}} + \sum_{\bar{\imath} \in (H^*)^n} \frac{p(i) - p_i}{\sum_{j \in \mathcal{V} \setminus H^*}(p(j) - p_j)} \cdot r(\bar{\imath}) \tag{126}$$

$$= p'_i + \frac{p(i) - p_i}{\sum_{j \in \mathcal{V} \setminus H^*}(p(j) - p_j)} \cdot \sum_{\bar{\imath} \in (H^*)^n} \left( p_{\text{draft}}(\bar{\imath}) - \sum_{j \in \text{set}(\bar{\imath})} S_{j,\bar{\imath}} \right) \tag{127}$$

$$= p'_i + \frac{p(i) - p_i}{\sum_{j \in \mathcal{V} \setminus H^*}(p(j) - p_j)} \cdot \left( \sum_{\bar{\imath} \in (H^*)^n} p_{\text{draft}}(\bar{\imath}) - \sum_{j \in H^*} p'_j \right) \tag{128}$$

$$= p'_i + \frac{p(i) - p_i}{-\psi(H^*)} \cdot \left( \sum_{j \in H^*} (p(j) - p'_j) - \psi(H^*) \right) \tag{129}$$

$$= p(i) + (p'_i - p_i) + \frac{p(i) - p_i}{-\psi(H^*)} \cdot \sum_{j \in H^*} (p(j) - p'_j). \tag{130}$$

In the fourth equality, we used the equality condition of Theorem 6.1, and the definition of $\psi$. Since $\psi(H^*) < 0$ and $p_i \le p(i)$, this allows us to bound the difference to $p(i)$ for $i \notin H^*$ as

$$\left| \sum_{\bar{\imath} \in \mathcal{V}^n} C_{i,\bar{\imath}} - p(i) \right| \le |p'_i - p_i| + \frac{p(i) - p_i}{-\psi(H^*)} \cdot \sum_{j \in H^*} |p'_j - p(j)| \le |p'_i - p_i| + \frac{p(i) - p_i}{-\psi(H^*)} \cdot \beta. \tag{131}$$

Hence, summing over all $i$ gives

$$\sum_{i \in \mathcal{V}} \left| \sum_{\bar{\imath} \in \mathcal{V}^n} C_{i,\bar{\imath}} - p(i) \right| \le \sum_{i \in H^*} |p'_i - p(i)| + \sum_{i \in \mathcal{V} \setminus H^*} |p'_i - p_i| + \sum_{i \in \mathcal{V} \setminus H^*} \frac{p(i) - p_i}{-\psi(H^*)} \cdot \beta \tag{132}$$

$$\le \beta + \alpha + \beta = \alpha + 2\beta. \tag{133}$$

In fact, each $C_{i,\bar{\imath}}$ is nonnegative as all $S_{i,\bar{\imath}}$ are nonnegative, and each $r(\bar{\imath}) \ge 0$ (by the inequality constraint of the inner system, which is exactly satisfied even for the truncated solver) and $p(i) \ge p_i$. Therefore, $C_{i,\bar{\imath}}$ represents a solution to the OTLP with total deviation at most $\alpha + 2\beta$ from the $p(i)$ equality constraints. Finally, the acceptance, i.e. the objective value in the OTLP, is

$$\sum_{\bar{\imath} \in \mathcal{V}^n} \sum_{i \in \text{set}(\bar{\imath})} C_{i,\bar{\imath}} = \sum_{\bar{\imath} \in \mathcal{V}^n} \sum_{i \in \text{set}(\bar{\imath})} S_{i,\bar{\imath}} = \sum_{i \in \mathcal{V}} p'_i. \tag{134}$$

By the Triangle Inequality, the fact that

$$\sum_{i \in H^*} p(i) + \sum_{i \in \mathcal{V} \setminus H^*} p_i = 1 + \sum_{i \in \mathcal{V} \setminus H^*} (p_i - p(i)) = 1 + \psi(H^*) = \alpha^* \tag{135}$$

is the optimal objective in the OTLP, and the fact that $\sum_{i \in H^*} |p'_i - p(i)| \le \beta$ and $\sum_{i \in \mathcal{V} \setminus H^*} |p'_i - p_i| \le \alpha$, this new objective value deviates by at most $\alpha + \beta$ from the optimal acceptance, as desired. $\qquad\square$

## M  EXPERIMENTAL SETUP

We run all experiments on Paperspace machines. For Llama-3 70/8B, we use a setup with 4xA6000 and an Intel Xeon Gold 5315Y CPU (32 cores, 3.20 GHz). For Gemma-2 27/2B, we use a setup with an A100-80GB and an Intel Xeon Gold 6342 CPU (12 cores, 2.80 GHz). We use temperature 1 sampling from the target and draft models in all settings.

Our data consists of 500 random eval prompts from GSM8K, HumanEval (only 164 total), IfEval, PopQA, and WildJailbreak, five diverse benchmarks spanning math reasoning, coding, instruction following, knowledge retrieval, and safety (Chen et al., 2021; Zhou et al., 2023a; Cobbe et al., 2021; Shen et al., 2024; Mallen et al., 2022). We take prompts only from the eval split to avoid test set contamination. For acceptance experiments, we compute token-averaged acceptances per dataset, and perform a simple average across datasets. For solve time experiments, as times are fairly prompt-agnostic, we select 40 random prompts from each dataset.

For our solve time experiments, we use the HiGHS LP solver (Huangfu & Hall, 2018) in SciPy's optimization module (Virtanen et al., 2020). We also use graph-tools (Peixoto, 2014), a heavily optimized Python network analysis package with a C++ backend and OpenMP support, to solve max-flow. To solve the convex minimization problem, we use the L-BFGS-B solver from SciPy (Virtanen et al., 2020). All solvers are only run on the CPU with numpy arrays, to minimize the impact of GPU setup differences on solve times.

## N  TOP-$k$ SAMPLING FOR THE RELAXED OTLP

Here, we formally describe how top-$k$ sampling of the draft model $q$ impacts the relaxed OTLP when $p_{\text{draft}}$ is formed by i.i.d. sampling $q$. Suppose that the top $k$ tokens of $q$ form $\mathcal{V}_0 \subseteq \mathcal{V}$. This means $p_{\text{draft}}(\bar{i}) = 0$ whenever $\bar{i} \notin \mathcal{V}_0^n$. Now, recall the relaxed OTLP is

$$\max_{\boldsymbol{S} \succeq 0} \sum_{i \in \mathcal{V}} \sum_{\bar{i} \in \mathcal{V}^n} S_{i,\bar{i}} \quad \text{s.t.} \quad \sum_{\bar{i} \in \mathcal{V}^n} S_{i,\bar{i}} \leq p(i) \quad \forall i \in \mathcal{V}, \quad \sum_{i \in \mathcal{V}} S_{i,\bar{i}} \leq p_{\text{draft}}(\bar{i}) \quad \forall \bar{i} \in \mathcal{V}^n, \tag{136}$$
$$S_{i,\bar{i}} = 0 \quad \forall i \in \mathcal{V}, \bar{i} \notin A_i.$$

For any $i \in V, \bar{i} \notin \mathcal{V}_0^n$, the $p_{\text{draft}}(\bar{i}) = 0$ upper bound constraint and nonnegativity of variables forces $S_{i,\bar{i}} = 0$. Furthermore, for any $i \notin \mathcal{V}_0, \bar{i} \in \mathcal{V}_0^n$, because $\bar{i} \notin A_i$ (it cannot contain $i$ as it has all elements in $\mathcal{V}_0$), the zero equality constraint above gives $S_{i,\bar{i}} = 0$. This means the only nonzero variables are those where $i \in \mathcal{V}_0, \bar{i} \in \mathcal{V}_0^n$. Plugging these into the above, simplifying the objective, and eliminating trivial constraints and variables we know to be zero, we get the following LP:

$$\max_{\boldsymbol{S} \succeq 0} \sum_{i \in \mathcal{V}_0} \sum_{\bar{i} \in \mathcal{V}_0^n} S_{i,\bar{i}} \quad \text{s.t.} \quad \sum_{\bar{i} \in \mathcal{V}_0^n} S_{i,\bar{i}} \leq p(i) \quad \forall i \in \mathcal{V}_0, \quad \sum_{i \in \mathcal{V}} S_{i,\bar{i}} \leq p_{\text{draft}}(\bar{i}) \quad \forall \bar{i} \in \mathcal{V}_0^n, \tag{137}$$
$$S_{i,\bar{i}} = 0 \quad \forall i \in \mathcal{V}_0, \bar{i} \in \mathcal{V}_0^n, i \in \text{set}(\bar{i}).$$

**This is exactly the same as the original relaxed OTLP if $\mathcal{V}$ was replaced by $\mathcal{V}_0$, and $p$ and $p_{\text{draft}}$ were replaced by their slices along $\mathcal{V}_0$ and $\mathcal{V}_0^n$.** Thus, top-$k$ sampling essentially reduces the problem to a smaller version of the relaxed OTLP, with $k^{n+1}$ parameters rather than $V^{n+1}$.

## O  GREEDY VS. I.I.D. ACCEPTANCE RATES

In Figure 2 and Figure 3, we compare acceptance rates for the i.i.d. and greedy $p_{\text{draft}}$ construction across Gemma-2 and Llama-3. We find i.i.d. performs worse than greedy for $k = 10$, but does better for all $k \geq 100$. In particular, it is nearly 2% better for most $n \geq 4$ in the latter case. Interestingly, the i.i.d. advantage seems to decrease as $n$ increases for higher $k$ in Llama-3, but not in Gemma-2. Exploring these trends is an interesting avenue for future work.

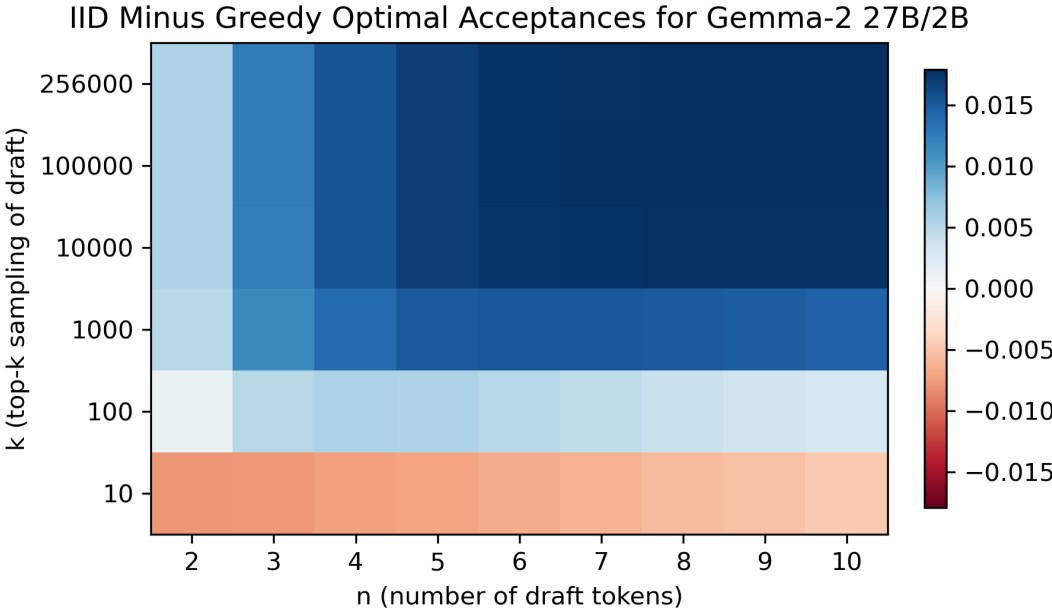

Figure 2: Comparison of i.i.d. versus greedy acceptance rates for Gemma-2 27B/2B across various choices of $n$ and top-$k$ sampling of the draft.

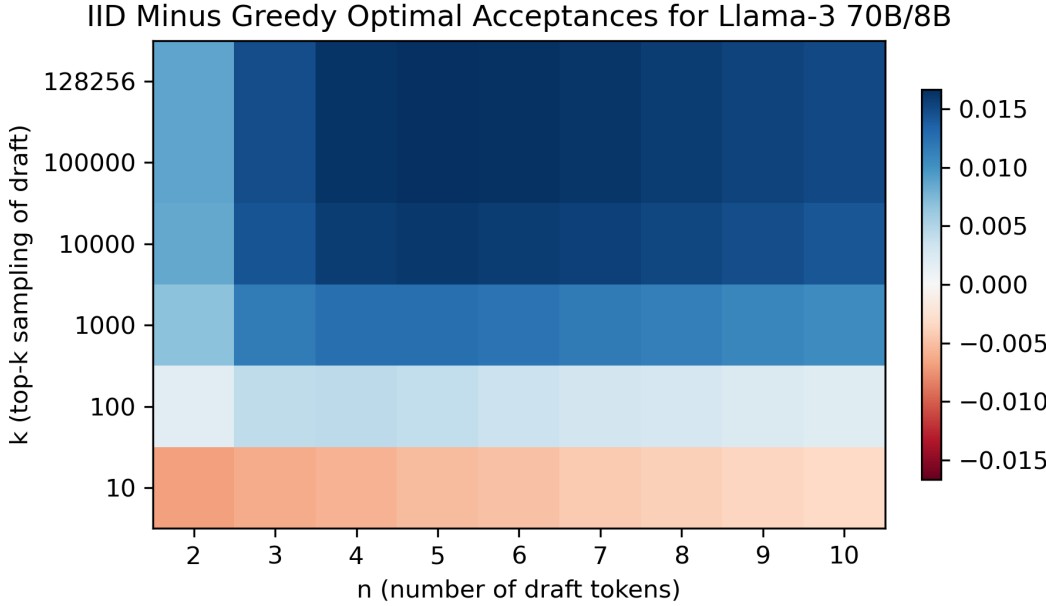

Figure 3: Comparison of i.i.d. versus greedy acceptance rates for Llama-3 70B/8B across various choices of $n$ and top-$k$ sampling of the draft.

# P  GEMMA-2 SOLVE TIME COMPARISON

Here, in Table 3, we compare the four methods' solver times for Gemma-2. Again, global resolution can hundreds of thousands times faster than the other methods. We also observe global resolution solve times for Gemma-2 are generally larger than those for Llama-3 (Table 1), but it is not clear how much of this is due to environment setup differences (CPU-only).

| $(k, n)$ | General LP | Max-Flow | Opt. Max-Flow | G.R. ($\tau = 10^{-3}$) | G.R. ($\tau = 10^{-4}$) |
|:---:|:---:|:---:|:---:|:---:|:---:|
| $(10, 2)$ | 7.85 | **3.18** | 3.26 | 7.46 (98.87%) | 10.63 (92.75%) |
| $(10, 3)$ | 59.43 | **5.05** | 5.10 | 24.64 (98.54%) | 33.95 (88.06%) |
| $(10, 4)$ | 4000+ | 95.73 | 94.21 | **57.21** (98.31%) | 77.42 (86.21%) |
| $(10, 5)$ | 400000+ | 10000+ | 10000+ | **98.10** (97.77%) | 130.68 (85.36%) |
| $(100, 2)$ | 4000+ | 95.13 | 82.94 | 49.58 (36.64%) | **43.49** (18.58%) |
| $(100, 3)$ | 400000+ | 200000+ | 200000+ | 107.65 (19.53%) | **32.81** (8.18%) |
| $(1000, 2)$ | OOM | 400000+ | 400000+ | **60.05** (27.5%) | 141.25 (8.04%) |

Table 3: Average Gemma-2 solve times (ms/token) over $k, n$, for the five i.i.d. OTLP solvers. General LP and max-flow are baselines, and optimized max-flow and global resolution ($\tau = 10^{-3}, 10^{-4}$) are ours. Lower numbers are better. Red numbers are lower bounds from small scale tests due to excessive runtime. Global resolution can be 10,000+ times faster than others. Global resolution deviates from the target distribution by at most $15\tau$ in L1 distance, and from optimal acceptance by at most $10\tau$. We include success rates for global resolution as it can terminate early sometimes.

# Q  TEMPERATURE ABLATIONS

In this section, we extend our results to different settings of target model sampling. We use Gemma-27B/2B with the same aggregate dataset as our acceptance rate experiments, and test temperature $0.2, 0.4, 0.6, 0.8$ sampling from the target model. We do not change the temperature of the draft distribution, so it remains $1.0$. Again, we test the effects of top-$k$ draft sampling.

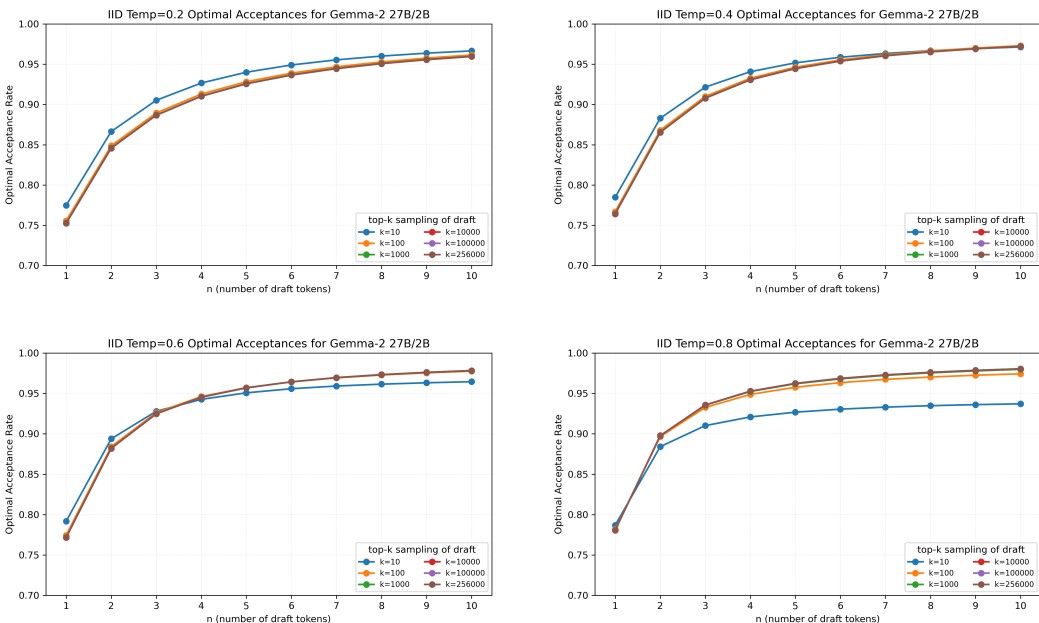

Figure 4: Optimal acceptance rates from $n$ i.i.d drafts with top-$k$ sampling with $n$ with the target/draft pair Gemma-2 27B/2B, for various target temperature settings $(0.2, 0.4, 0.6, 0.8)$. Until temperature $0.8$, increasing $k$ past 10 results in little acceptance gains for reasonable values of $n$.

We begin by plotting the acceptance rates in Figure 4. We observe that for temperatures $0.2, 0.4$, the $k = 10$ setting achieves essentially the highest acceptance rate for all values of $n$: increasing $k$ does not lead to any noticeable acceptance gains. For temperature $0.6$, $k = 10$ performs better until around $n = 4$, but afterwards $k \geq 100$ performs better (although the difference is not significant). For temperature $0.8$, $k = 10$ is significantly worse than other settings as $n$ increases, although $k = 100$ has comparable acceptances to $k \geq 1000$. These results demonstrate that decreasing the temperature allows top-$k$ sampling to achieve extremely high acceptance rates for small values of $k$, thereby making approaches which staunchly rely on it (e.g. LP and max-flow solvers) feasible and effective.

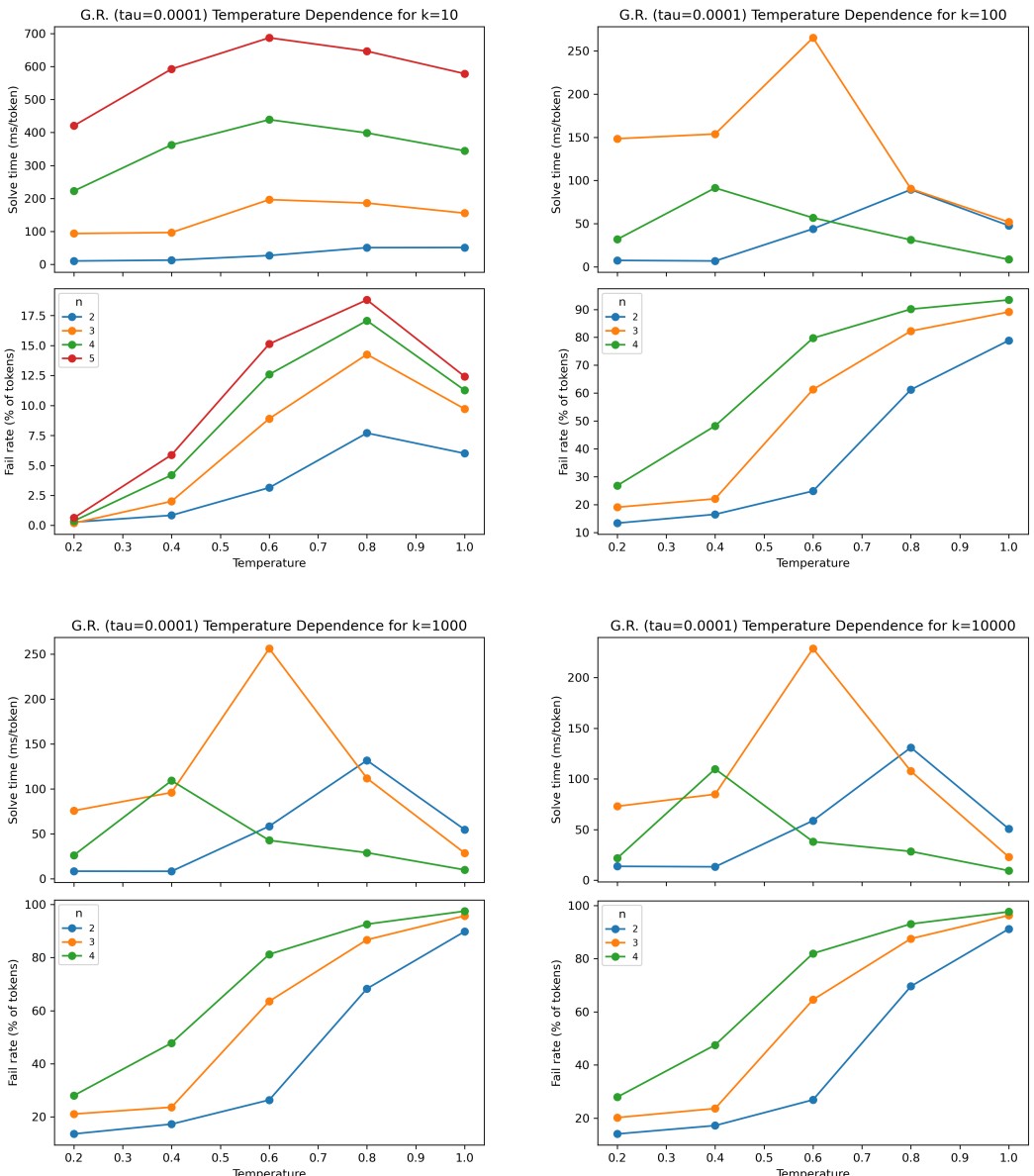

Figure 5: Solve times and failure rates of global resolution with $\tau = 10^{-4}$ (Gemma-2 27B/2B) for choices of $k \in \{10, 100, 1000, 10000\}$ and $n \in \{2, 3, 4, 5\}$, plotted over increasing temperature.

We also examine the efficacy of global resolution with $\tau = 10^{-4}$ as temperature varies in Figure 5. For $k > 10$ (not the top left graphs), failure rates increase significantly with temperature: global resolution is most applicable at low temperature settings. Also, for all $k$ and $n$, global resolution solve times increase until a certain temperature, and then decrease thereafter. Given the high success rate and low solve times at lower temperatures, global resolution shows great promise in this regime.

# R    MULTI-STEP EXPERIMENTS

Here, we implement global resolution in the framework SpecTr (Sun et al., 2023) to measure end-to-end latency in multi-step speculative decoding systems. SpecTr first drafts $K$ i.i.d. paths of length $L$ autoregressively from the draft model to form a draft tree, and then incrementally uses any OTLP solver (with $n$ being set to the number of child nodes) to advance from the root of the tree to child nodes, until it lands off the tree. We use global resolution for our OTLP solver with $k = 100$ for top-$k$ draft sampling, and when it fails, we resort to standard sampling from $p$. We run our experiments on Gemma-27B/2B with temperature 1.0 target sampling on the same dataset as our solve time experiments. Our results for $K = 2, 3, 4$ and $L = 8$ are shown in Table 4.

| $K$ | Block efficiency (tokens/$M_p$ call) | Walltime speedup (relative to vanilla) |
|---|---|---|
| 2 | 2.23 | 1.84 |
| 3 | 2.54 | 1.89 |
| 4 | 2.65 | 1.98 |

Table 4: Block efficiency (number of decoded tokens per call to target model) in and walltime speedup (over vanilla autoregressive decoding) for SpecTr with global resolution under different values of $K$ (number of i.i.d. paths). Experiments are run on Gemma-27B/2B with $L = 8$.

Our walltime speedups in the best setting ($K = 4$) are nearly $2\times$ better than vanilla decoding. We also emphasize that these numbers are highly conservative estimates. Unlike previous OTLP solvers used in the SpecTr framework, global resolution is situational: it is provably nearly optimal, but cannot always be used. This means the walltime and efficiency numbers above can be improved significantly by using other OTLP solvers like K-SEQ (Sun et al., 2023) or vocabulary truncation (Khisti et al., 2025) when global resolution fails, rather than our simple $p$-sampling heuristic (which is a valid OTLP solver, but has poor acceptances). Furthermore, by modifying the choice of $k$ in top-$k$ draft sampling, one might be able to use global resolution to get high acceptances in failure scenarios for other $k$. We leave such selection of OTLP solvers and their parameters to future work.

We also theoretically examine the properties of error compounding in multi-step setups. We observe that the multi-step error scales approximately linearly with the draft block length $L$.

**Theorem R.1** (Multi-Step Error). *Let $p_L(\cdot)$ denote the distribution of the next $L$ tokens, under autoregressive decoding from the target model. Let $\widehat{p_L}(\cdot)$ denote the same joint distribution, but now under decoding by global resolution with error tolerance $\tau$. Then $\|p_L - \widehat{p_L}\|_1 \le 15L\tau$.*

*Proof.* For each $j = 0, \ldots, L$, define $p_L^{(j)}$ to be the distribution of the next $L$ tokens, under autoregressive sampling from the target model for the first $j$ tokens and then decoding by global resolution with tolerance $\tau$ thereafter. We use the shorthand notation $a_{1:L}$ to denote a sequence of $L$ tokens $(a_1, \ldots, a_L)$ in $\mathcal{V}^L$, with $a_{1:i}$ denote the slice of the first $i$ tokens and $a_{1:0} = \emptyset$. By applying the Triangle Inequality, it suffices to show for each $j = 0, \ldots, L - 1$ that

$$\left\| p_L^{(j+1)} - p_L^{(j)} \right\|_1 = \sum_{a_{1:L} \in \mathcal{V}^L} \left| p_L^{(j+1)}(a_{1:L}) - p_L^{(j)}(a_{1:L}) \right| \le 15\tau, \tag{138}$$

as then adding up these $L$ inequalities gives the desired upper bound of $15\tau L$.

To prove this, observe for any $a_{1:L}$ that

$$\left| p_L^{(j+1)}(a_{1:L}) - p_L^{(j)}(a_{1:L}) \right| \tag{139}$$

$$= \left| \prod_{i=0}^{j} p(a_{i+1}|a_{1:i}) \prod_{i=j+1}^{L-1} \widehat{p}(a_{i+1}|a_{1:i}) - \prod_{i=0}^{j-1} p(a_{i+1}|a_{1:i}) \prod_{i=j}^{L-1} \widehat{p}(a_{i+1}|a_{1:i}) \right| \tag{140}$$

$$= \left( \prod_{i=0}^{j-1} p(a_{i+1}|a_{1:i}) \prod_{i=j+1}^{L-1} \widehat{p}(a_{i+1}|a_{1:i}) \right) \cdot \left| \widehat{p}(a_{j+1}|a_{1:j}) - p(a_{j+1}|a_{1:j}) \right|, \tag{141}$$

where $p(\cdot|\cdot)$ and $\widehat{p}(\cdot|\cdot)$ are the conditional next-token distributions that induce the joint distributions $p_L, \widehat{p_L}$. By the global resolution approximation guarantee on L1 target distribution error at the end of Section 6.3, we see that each $\|p(\cdot|a_{1:i}) - \widehat{p}(\cdot|a_{1:i})\|_1 \leq 15\tau$. Therefore, summing the above bound over all $a_{1:L} \in \mathcal{V}^L$ gives the desired bound. Below, our sole inequality uses the L1 approximation bound, and the third line cancels out the summation of $\widehat{p}$ products over $a_{j+2:L}$ to one.

$$\sum_{a_{1:L} \in \mathcal{V}^L} \left| p_L^{(j+1)}(a_{1:L}) - p_L^{(j)}(a_{1:L}) \right| \tag{142}$$

$$= \sum_{a_{1:L} \in \mathcal{V}^L} \left( \prod_{i=0}^{j-1} p(a_{i+1}|a_{1:i}) \prod_{i=j+1}^{L-1} \widehat{p}(a_{i+1}|a_{1:i}) \right) \cdot |\widehat{p}(a_{j+1}|a_{1:j}) - p(a_{j+1}|a_{1:j})| \tag{143}$$

$$= \sum_{a_{1:j+1} \in \mathcal{V}^{j+1}} \left( \prod_{i=0}^{j-1} p(a_{i+1}|a_{1:i}) \right) \cdot |\widehat{p}(a_{j+1}|a_{1:j}) - p(a_{j+1}|a_{1:j})| \tag{144}$$

$$= \sum_{a_{1:j} \in \mathcal{V}^{j+1}} \left( \prod_{i=0}^{j-1} p(a_{i+1}|a_{1:i}) \cdot \left( \sum_{a_{j+1} \in \mathcal{V}} |\widehat{p}(a_{j+1}|a_{1:j}) - p(a_{j+1}|a_{1:j})| \right) \right) \tag{145}$$

$$\leq \sum_{a_{1:j} \in \mathcal{V}^{j+1}} 15\tau \prod_{i=0}^{j-1} p(a_{i+1}|a_{1:i}) = 15\tau \sum_{a_{1:j} \in \mathcal{V}^{j+1}} \prod_{i=0}^{j-1} p(a_{i+1}|a_{1:i}) = 15\tau. \tag{146}$$

This completes the proof of the multi-step compounding error. $\qquad\square$

## S    GENERALIZING TO NON-IID DRAFTS

Here, we describe the potential of extending global resolution to non-i.i.d. drafting frameworks. We consider three regimes: sampling from a single distribution without replacement, sampling $n = 2$ independent non-identical draft models, and sampling $n \geq 2$ independent non-identical draft models. In Table 5, we examine the three-step breakdown of global resolution from the beginning of Section 6. For each of the three regimes, we classify its extension to each of the three steps under **Yes** (extension is immediate), **Possible** (extension is likely possible but requires additional work), and **No** (fundamental roadblock prevents the approach from going through).

| Drafting | Step 1 | Step 2 | Step 3 |
|---|---|---|---|
| Sampling without replacement | Yes | Yes | Possible |
| $n = 2$ independent and distinct | Yes | Possible | Possible |
| $n \geq 3$ independent and distinct | No | No | Possible |

Table 5: Extending global resolution to three non-i.i.d. drafting regimes. Yes means the extension is immediate; Possible means it requires some work; No means there is a major obstacle.

We first cover step 1: the computation of $H^*$. In sampling without replacement, this is straightforward by following the $O(V \log V + nV)$ algorithm in Theorem 4 and Section 4.2.2 of Hu et al. (2025). For $n = 2$ distinct independent drafts, this is now possible following our reduction to QBPO in Appendix D. However, for $n \geq 3$ distinct independent drafts, this QBPO reduction no longer holds. While our work in Appendix C shows this is still an instance of submodular minimization, the most efficient algorithms (Iwata, 2008) would require at least $O(V^5 \log V)$ work, which is infeasible for most vocabularies. Approximating $H^*$ is possible, but this is a totally separate direction of work, and ensuring that these $H^*$ approximations give a strong bound on the final error may be difficult.

Now, we cover step 2: the calculation of outer residuals $p_i$. The key idea here is that the construction of the outer residual LP (Appendix G) and the greedy polymatroid algorithm to get $p_i$ (Appendix O) only rely on the fact that $\psi$ is submodular, and then we can compute $\min \psi$ terms and take consecutive differences to get $p_i$. For sampling without replacement, as in the above paragraph, the algorithm from Hu et al. (2025) facilitates the same approach and even allows the $O(V \log V)$ optimization (cumulative minimums) at the end of Section 6.1 to hold. For $n = 2$ distinct independent drafts, the only roadblock is that computing consecutive differences of $\min \psi$ terms may be costly without an

approach like cumulative minimums, so some additional work is required on this front. For $n \geq 3$ distinct independent drafts, the runtime concerns in the above paragraph again prevent us from even computing a single $\min \psi$ term, so this again appears an intractable roadblock.

Finally, we cover step 3: the convex minimization approach. On their own, nothing from Theorem 6.4 and Theorem 6.5 inherently relies on the structure of the drafts, once $p_i$ and $H^*$ are solved for in steps 1 and 2. The only dependence on drafting comes in computing provable error bounds and selecting the truncation set $T$. For all three drafting regimes, given that there is a simple form for the sum of $p_{\text{draft}}$ over Cartesian products of sets (which are used in tunable error bounds), it seems highly likely that this approach could work with some modifications.

To summarize, sampling without replacement seems almost immediately tractable for global resolution, and so does $n = 2$ independent distinct drafts with some additional work on outer residual computations. However, $n \geq 3$ independent distinct drafts seems intractable due to the higher-degree submodular minimization bottleneck, and would likely require a different approach.

We also note that while global resolution requires all three of these steps to go through, our other baseline solvers do not. General LP and max-flow solves require none of these steps, and optimized max-flow only requires step 1 (and none of these assume any information about the structure of $p_{\text{draft}}$). Thus, in all these drafting regimes, these methods can be used after top-$k$ draft sampling reduces the OTLP size.

