# OpenReview forum: "Global Resolution: Optimal Multi-Draft Speculative Sampling via Convex Optimization"
_ICLR.cc/2026/Conference — ICLR 2026 Oral_

### Official Review · Reviewer_TpbB · 2025-10-30

**Soundness:** 3
**Presentation:** 3
**Contribution:** 3
**Rating:** 6
**Confidence:** 3

**Summary:**

The paper studies multi-draft speculative sampling for autoregressive decoding. It (i) proves that recent importance-sampling and subset-selection formulations are equivalent to an exponentially large relaxed OTLP, which is still intractable, (ii) reverse-engineer subset selection to a max-flow formulation, and (iii) apply polymatroid theory to reduce the problem to a convex program for i.i.d. drafts from a single draft model. Theoretical guarantees are given and empirical results demonstrate the effectiveness of the proposed method.

**Strengths:**

- The paper turns an exponential OTLP into a convex program in tolerable number of variables (under i.i.d. drafts) is a substantial conceptual and practical advance.
- It also provides approximation error guarantees, which helps to determine the time-precision trade-off.
- The experiments demonstrated the strong performance of the proposed method.

**Weaknesses:**

- The main convex reduction requires i.i.d. drafts from a single $q$. There are some works adopt mixture drafts across experts. The iid configuration may be one concern of the proposed method.
- The analysis focuses on solver runtime rather than full end-to-end throughput.

**Questions:**

- When extending to the multi-step case, can the authors comment on how the compounding error is expected to scale?

---

> ### Author Response · Authors · 2025-11-19
> **Rebuttal by Authors**
>
> Thank you for your detailed coverage of our work! We appreciate that you found our paper makes a substantial advance, and that our approach delivers strong performance improvements.
>
> We address your other points below:
>
> **W1**
>
> Thank you for raising this valid point on the limitation to i.i.d. drafts.
>
> We have updated our manuscript to provide further details on extensions to other drafting methods in Appendix S. Our approach naturally makes significant progress towards independent but non-identical drafts, as well as sampling without replacement.
>
> Global resolution is a three-step process: (1) solve for $H^\star$ efficiently via methods from Hu et. al., (2) solve for the outer residuals $p_i$ with polymatroid theory, and (3) solve the convex problems in Theorems 6.4 and 6.5. Another approach we consider, optimized max-flow, only requires step (1) and then directly solves the complementary slackness system. All other approaches (general LP, max-flow) in our paper require none of these steps.
>
> While we did not originally mention this in our manuscript due to space limitations and a focus on the i.i.d. setting, the methods from Hu et. al. also allow efficient computation of $H^\star$ in the sampling without replacement regime (see Theorem 4 and Section 4.2.2 of their paper). The only difference is that the runtime of step (1) is $O(V \log V + nV)$, rather than $O(V \log V)$.
>
> This has two implications. First, all approaches that we consider other than global resolution (general LP, max-flow, optimized max-flow) can be directly applied to the sampling without replacement regime, with similar theoretical complexity. Second, the only roadblock to extending global resolution to this setting is to generalize steps (2) and (3). This is tractable following a similar greedy polymatroid and convex minimization approach (the function $\psi$ is still submodular for sampling without replacement), and we leave the exact details to future work.
>
> For the extension to independent but not identical drafts, we have already provided some initial discussion in our manuscript. For example, in Section 5.1, we state that in the 3-step process of global resolution mentioned under W1 above, the step (1) of solving for $H^\star$ can be done efficiently using QBPO for $n=2$ independent (but distinct) drafts, and $\psi$ is still submodular, and so extending global resolution (and our other evaluated methods here) here is similar to extending it to sampling without replacement.
>
> For larger $n$, $\psi$ is still submodular. This means general submodular minimization works to solve for $H^\star$, but for larger vocabularies this can be intractable. Thus, future work could explore approximations to $H^\star$ that allow the rest of global resolution to follow through.
>
> **W2**
>
> We agree that it is important to measure full end-to-end throughput when evaluating speculative decoding methods.
>
> Following your feedback, we implemented global resolution in the multi-step framework SpecTr [https://arxiv.org/pdf/2310.15141], a standard speculative decoding algorithm which can be run with any OTLP solver. Our manuscript (see Appendix R) now includes block efficiency (theoretical speedup) and walltime (real speedup) numbers for SpecTr with global resolution, to accurately measure the throughput improvements from global resolution: see our global comment to reviewers for more details (https://openreview.net/forum?id=gpsczXOsHn&noteId=a56AUrvD1D).
>
> **Q1**
>
> We have included a theoretical analysis of compounding error in the multi-step setting in Appendix R. The error scales sublinearly with the length of the draft block. If we run global resolution at every step with error threshold $\tau=0.0001$ and draft block length $L=8$, the effective L1 error in the next-$L$-token distribution will provably remain under $15L\tau=0.012$.
>
>
> Again, thank you for your detailed feedback and suggestions on our work! If you have any lingering questions or comments on our response, or other suggestions on how to improve our work, please let us know. We are glad to continue discussion at any time until the rebuttal period ends. We have made a significant effort to incorporate your experimental feedback and drafting generalizations in our work, and would appreciate it if you would consider raising your score in light of our response. Thank you!

---

### Official Review · Reviewer_YzHd · 2025-11-01

**Soundness:** 4
**Presentation:** 4
**Contribution:** 4
**Rating:** 8
**Confidence:** 3

**Summary:**

This paper addresses the computational intractability of optimal multi-draft speculative sampling. Previous work either computed optimal acceptance analytically (for restricted cases) or estimated it without recovering the optimal transport (OT) itself. The authors prove the equivalence of prior formulations (subset selection and canonical decomposition) and introduce Global Resolution, a convex-optimization-based solver that achieves near-optimal OT in the i.i.d. draft setting. The method reduces an exponentially large LP to a convex problem, with guaranteed deviation and practical runtimes (< 100 ms/token).

**Strengths:**

Important problem. Tackles a key bottleneck in speculative decoding efficiency for LLM inference.

Theoretical soundness. Clear equivalence proofs (Sec. 4), a max‑flow reduction with complementary slackness (Sec. 5), and convex programs for inner/outer systems with explicit error bounds (Theorems 6.4–6.5, Lemma 6.6).

Practical algorithm. Global Resolution achieves < 100 ms/token OT‑solve time while maintaining near‑optimal acceptance. This is a substantial empirical improvement over generic LP/max‑flow baselines in the tested regime.

**Weaknesses:**

Not a major weakness, but the current method is limited to i.i.d. drafts. In practice, sampling without replacement (i.e., enforcing distinct drafts) typically yields better performance. Extending the approach to that regime, as well as to multi‑step setups, is left for future work. However, overall, the paper already makes great progress.

Typos:
Theorem 4.6, Equation (17): missing a summation on the LHS.
Equations (22), (23): the conditional distributions are omitted in the notation.

**Questions:**

Do Global Resolution extends (with guarantees) to independent but non‑identical drafts or sampling without replacement drafts?

---

> ### Author Response · Authors · 2025-11-19
> **Rebuttal by Authors**
>
> Thank you for your strong review of our work! We appreciate that you find that our work tackles an important problem, is theoretically sound, and achieves a substantial empirical improvement.
>
> We address your other points below:
>
> **W1**
>
> Thank you for raising this point. We agree that extending global resolution to sampling without replacement and multi-step setups is a natural direction for future work. Our approach naturally leads to significant progress in this direction, and we have updated our transcript to clarify this in Appendix S.
>
> To briefly summarize, global resolution involves (1) solving for $H^\star$ efficiently via methods from Hu et. al., (2) solving for the outer residuals $p_i$ with polymatroid theory, and (3) solving the convex problems in Theorems 6.4 and 6.5. Another approach we consider, optimized max-flow, only requires step (1) and then directly solves the complementary slackness system. All other approaches (general LP, max-flow) we consider require none of these steps.
>
> While we did not originally mention this in our manuscript due to space limitations and a focus on the i.i.d. setting, the methods from Hu et. al. also allow efficient computation of $H^\star$ in the sampling without replacement regime (see Theorem 4 and Section 4.2.2 of their paper). The only difference is that the runtime of step (1) is $O(V \log V + nV)$, rather than $O(V \log V)$.
>
> This has two implications. First, all approaches that we consider other than global resolution (general LP, max-flow, optimized max-flow) can be directly applied to the sampling without replacement regime, with similar theoretical complexity. Second, the only roadblock to extending global resolution to this setting is to generalize steps (2) and (3). This is tractable following a similar greedy polymatroid and convex minimization approach (the function $\psi$ is still submodular for sampling without replacement), and we leave the exact details to future work.
>
> For the extension to multi-step, we have implemented global resolution in the multi-step framework SpecTr (https://arxiv.org/abs/2310.15141) to measure real-world latency. See our global comment for more details (https://openreview.net/forum?id=gpsczXOsHn&noteId=a56AUrvD1D).
>
> **W2**
>
> Thank you for noticing the typos in Equations (22) and (23)! We have updated our proof in Appendix B to use the same conditional distribution notation in Theorem 4.1.
>
> We wanted to clarify your point on Theorem 4.6, Equation (17). Since we do not have a Theorem 4.6, were you referring to Theorem 6.5, Equation (17)? We believe there is no missing summation on the left hand side there.
>
> **Q1**
>
> Under W1, we discussed how to potentially extend this work to sampling without replacement.
>
> For the extension to independent but not identical drafts, we have already provided some initial discussion in our manuscript. For example, in Section 5.1, we state that in the 3-step process of global resolution mentioned under W1 above, the step (1) of solving for $H^\star$ can be done efficiently using QBPO for $n=2$ independent (but distinct) drafts, and $\psi$ is still submodular, and so extending global resolution (and our other evaluated methods here) here is similar to extending it to sampling without replacement.
>
> For larger $n$, $\psi$ is still submodular. This means general submodular minimization works to solve for $H^\star$, but for larger vocabularies this can be intractable. Thus, future work could explore approximations to $H^\star$ that allow the rest of global resolution to follow through.
>
>
> Thank you for your detailed review of our work! Please let us know if you have any comments or questions about any part of our response, or any further ideas on how to improve  our work. We are happy to continue discussion at any time until the discussion period ends. We very much appreciate the high score you have given our submission; if you feel that the above addresses your questions comprehensively, we would kindly request if you would consider raising it even further. Thank you!

---

### Official Review · Reviewer_ZwDM · 2025-11-03

**Soundness:** 3
**Presentation:** 3
**Contribution:** 3
**Rating:** 6
**Confidence:** 3

**Summary:**

This paper addresses the inefficiency of *multi-draft speculative decoding* for large language models (LLMs), where computing the optimal transport (OT) verification criterion requires solving an exponentially large linear program (OTLP) over \(V^n\) variables. The authors first unify two prior theoretical formulations—importance sampling and subset selection—showing that both are equivalent to a relaxed exponential OTLP. They then reverse-engineer the subset-selection view and reformulate the problem as a max-flow optimization, which, via a novel application of polymatroid theory, is further reduced to a convex minimization problem with at most \(V\) variables.  This reduction yields a new algorithm called Global Resolution, which achieves provably optimal acceptance rates in the i.i.d. single-distribution setting where \(n\) draft tokens are sampled from the same draft model. Empirically, the paper measures acceptance rates and solver runtimes across different numbers of drafts \(n\) and top-\(k\) sampling configurations. Results show that the proposed solver can achieve over 90% acceptance with less than 100 ms overhead per token, and negligible deviation from the true target distribution.

**Strengths:**

The theoretical contribution of this paper is substantial. It successfully unifies two previously disjoint theoretical perspectives on speculative decoding—importance sampling and subset selection—into a single coherent framework. Building upon this, the authors further derive a novel convex minimization formulation via polymatroid theory, which drastically reduces the exponential complexity of the original OT linear program to a problem in at most \(V\) variables.

**Weaknesses:**

This paper does not discuss the robustness of the proposed algorithm with respect to temperature. Intuitively, different temperature values shape different draft and target distributions, which could significantly affect the optimal acceptance rate. It remains unclear how the proposed approach performs under varying temperature settings. Moreovere, this paper does not intergrate their algorithm in real-world speculative decoding systems to show how the algorithm/theory can improve the latency of LLM decoding.

**Questions:**

1. Can you give more details about how you construct the distribution in your experiments, e.g., tempearture.
2. Can you add some temperature experiments?

---

> ### Author Response · Authors · 2025-11-19
> **Rebuttal by Authors**
>
> Thank you for your detailed review! We appreciate that you found our theoretical unification of importance sampling and subset selection to be impactful, and that you found our proposed improvements to existing OTLP solvers to be substantial.
>
> We address your other points below:
>
> **W1**
>
> Thank you for raising the importance of temperature variation in our experiments. We agree that temperature can significantly affect the target and draft distributions, which impacts not only the optimal acceptance rate but also the solve time of global resolution.
>
> As such, we have run additional temperature experiments and updated our manuscript to include these. We first ran temperature-dependent acceptance rate experiments, and then ran solve time experiments, across temperatures 0.2, 0.4, 0.6, and 0.8 for global resolution with $\tau=10^{-4}$ on Gemma-27B/2B (the temperature 0.0 is not really relevant to OTLP solvers, since greedy decoding is deterministic). All results are shown in Appendix Q.
>
> For acceptance rate graphs (i.e. like our Figure 1 graph) for various temperatures, see the quick links below. We observe for lower temperatures (0.2, 0.4, 0.6) that increasing $k$ in top-$k$ sampling of the draft model offers essentially no benefit: $k=10$ ends with the highest acceptance rate for most reasonable values of $n$. This means solvers other than global resolution (which require $k$ to be small) have the potential to efficiently get high acceptance rates.
>
> Temperature 0.2: https://ibb.co/YFjSDpKx
>
> Temperature 0.4: https://ibb.co/PZ8LkKbw
>
> Temperature 0.6: https://ibb.co/p62WrSx4
>
> Temperature 0.8: https://ibb.co/DgjysmYX
>
> For graphs showing how global resolution solve time and success rate evolve with temperature for different top-$k$ draft sampling settings, see the quick links below. Here, we observe that the failure rate of global resolution significantly decreases in lower-temperature settings, with lower solve times as well, naturally leading to higher acceptance rates. This is not only true for the ideal $k=10$ setting mentioned in the acceptance rate experiments, but even for $k$ up to $10000$. **Thus, our original results actually frame global resolution in its worst-performing setting: decreasing the temperature improves the efficacy of our approach.**
>
> Top-10 sampling: https://ibb.co/7tgqqB6t
>
> Top-100 sampling: https://ibb.co/6cK76KWR
>
> Top-1000 sampling: https://ibb.co/TMgrXRqX
>
> Top-10000 sampling: https://ibb.co/FSTTzX4
>
> Intuitively, this makes sense as global resolution is particularly efficient when the target distribution is concentrated over a small subset of tokens, as is the case when the temperature decreases.
>
> **W2**
>
> We agree that global resolution should be evaluated in real-word speculative decoding systems.
>
> Following your feedback, we have implemented global resolution in the real-world framework SpecTr (https://arxiv.org/pdf/2310.15141), a standard multi-step speculative decoding algorithm which makes use of OTLP solvers. This is what some existing works use to make realistic latency evaluations. See our global comment to reviewers for more details (https://openreview.net/forum?id=gpsczXOsHn&noteId=a56AUrvD1D).
>
> **Q1**
>
> All our experiments involve temperature 1 sampling from the target model and draft model.
>
> **Q2**
>
> We have addressed the additional temperature experiments in W1 above.
>
>
> Thank you again for your comprehensive feedback and insightful suggestions. Please let us know if you have any further questions or comments on our responses. We are happy to continue discussion at any time for the remainder of the rebuttal period. We have made a significant effort to incorporate your experimental suggestions, and we would appreciate it if you would consider raising your score. Thank you!

---

### Official Review · Reviewer_UvVU · 2025-11-12

**Soundness:** 3
**Presentation:** 2
**Contribution:** 3
**Rating:** 6
**Confidence:** 2

**Summary:**

First, I apologize for the late review; I was invited to join late.

The paper studies the problem of speculative sampling with a draft sequence length of one (single-step), which can be formulated mathematically as an optimal transport (OT) problem. The paper studies when the optimal transport problem can be efficiently solved (which lets one hence compute the optimal acceptance ratio). In particular cases, the authors derive an efficient algorithm to solve the OT problem. Several experiments demonstrate the empirical efficacy of the proposed method in solving the OT problem vs other approaches, and also demonstrate the effect of changing various parameters on the resulting acceptance rates.

**Strengths:**

The paper establishes an algorithm for solving the optimal transport (OT) problem arising from speculative sampling. The algorithm has some compelling theoretical properties and has interesting connections with seemingly unrelated topics like max flow and linear programming. Experimental results comparing the resulting proposed algorithm vs other methods of solving the OT problem are convincing of the algorithm's efficacy in solving the OT problem compared to off-the-shelf solving methods. While I had some issues with the theory (see below), the experimental improvements combined with the proposed theory for solving the OT are convincing.

**Weaknesses:**

The main weakness of the work is as follows:
1. It would be interesting to see that the proposed algorithm improves on one way to demonstrate this improvement would be to demonstrate that the proposed algorithm is better empirically. However there is no empirical comparison to Hu et al (2025) or other papers that have studied this problem.

2. The proposed algorithm seems to require enumerating over $(H^{\star} \cup T)^n - (H^{\star})^n$ (see Theorem 6.4) (e.g. to compute gradients of $\Phi_T$). Yes $T$ can be chosen, but as $|T| \ge 1$ this still takes time at least $|H^{\star}|^{n-1} \cdot |T| \cdot n \ge |H^{\star}|^{n-1}$. As such, how is the proposed algorithms computationally efficient (which is the point of the paper), am I missing something? It would be great if the authors could clarify this point.

3. The writing could be significantly improved. It seems more conventional to present Theorems 6.4 and 6.5 to begin with, describe (at least at a high level) the algorithm, and then argue for its correctness. The current presentation does so backwards. It also took several reads for me to understand some of the key points of the paper, like how $H^{\star}$ can be obtained efficiently in $O(V \log V)$ time from Hu et al 2025 so therefore it is reasonable to assume knowledge of $H^{\star}$, and that the crucial point is to solve the OT rather than compute $\alpha^{\star}$. Overall the paper has interesting ideas but it was hard to read.

**Questions:**

Could the authors please clarify weakness 2 above, on the computational efficiency of the proposed algorithm?

It would be also nice if the authors could add some more references and discussion on why solving the single-step speculative sampling problem is important in practice.

---

> ### Author Response · Authors · 2025-11-19
> **Rebuttal by Authors**
>
> Thank you for your comprehensive review of our work! We appreciate that you found our theoretical approach to the OTLP interesting, and that you found our global resolution method to be empirically effective compared to existing solvers.
>
> We address your other points below:
>
> **W1**
>
> Thank you for raising this point. We do compare our method to Hu et. al. in Appendix O.
>
> As mentioned at the end of Page 8 after Figure 1, we compare the theoretical optimal i.i.d. acceptance rates (which global resolution achieves) to the greedy construction acceptance rates from Hu et. al. in Appendix O. We find that for all top-$k$ sampling settings with $k \geq 100$, global resolution can achieve nearly 2% higher acceptance rates on average.
>
> **W2**
>
> Thank you for raising this important point about the computational complexity of the convex minimizer in Theorem 6.4. We show how to overcome this dependence on $H^*$ in Appendix K.
>
> As you mention, it is true that the summation over $(H^\star \cup T)^n - (H^\star)^n$ is exponentially large in terms of $|H^\star|$. However, the summation is a linear combination of terms like $\log(e^{\alpha_{i_1}}+\ldots+e^{\alpha_{i_k}})$, where each of $i_1,\ldots,i_k$ lies in $T$ but not $H^\star$. This means that by grouping coefficients of common LogSumExp terms, which can be done efficiently using the Principle of Inclusion and Exclusion (PIE), we retain at most one term for each subset of $T$ with size at most $n$. This reduces the problem size from at least $|H^\star|^{n-1} \cdot |T| \cdot n$ to at most $|T|^n$, in fact roughly around $|T|^n/n!$. We also force global resolution to terminate early if $|T|^n/n!$ is too large, which is why our Table 1 results also include success rates for global resolution. This completely removes the dependence on $|H^*|$, and ensures the runtime of the convex minimization problem is negligible in practice.
>
> Thank you again for mentioning this point. We have now updated our manuscript to indicate the above more clearly, in addition to the details remaining in Appendix K.
>
> **W3**
>
> We agree with your points about the organization of our writing.
>
> Following your feedback, we have updated our manuscript. At the beginning of Section 6, we now include a brief overview of the overall algorithm and its decomposition into (1) solving for $H^\star$ efficiently via methods from Hu et. al., (2) solving for the outer residuals $p_i$ with polymatroid theory, and (3) solving the convex problems in Theorems 6.4 and 6.5. We have also emphasized the importance of solving the OT instead instead of just computing $\alpha^\star$, which only requires step (1) but does not result in a real verification algorithm.
>
> We did not move Theorems 6.4 and 6.5 to the beginning of the section because Theorem 6.4 implicitly makes use of the outer residuals $p_i$ that are solved for in Theorem 6.2 and Lemma 6.3 (which are technically parts of the full algorithm, not correctness theorems), so we feel that it may confuse readers. However, we have clarified this dependency in our outline at the beginning of this section.
>
> Thank you again for your detailed feedback, allowing us to improve the clarity of our submission.
>
> **Q1**
>
> We have clarified the computational efficiency weakness in the W2 section above.
>
> **Q2**
>
> In the last paragraph of our related works section, we now explain why solving the single-step speculative sampling problem, i.e. producing an approximate or exact OTLP solver, is important in practice. SpecTr (https://arxiv.org/pdf/2310.15141) provides a general framework to obtain a multi-step speculative decoding algorithm from a single-step OTLP solver: see Algorithm 3 of their paper, where line 1 represents the use of this solver. SpecTr uses K-SEQ as their approximate solver, and related works such as “Canonical Decomposition” (https://arxiv.org/pdf/2410.18234) and Hu et. al.’s “Towards Optimal Multi-Draft Speculative Decoding” (https://arxiv.org/pdf/2502.18779) use different OTLP solvers. Improving the acceptance rate of the OTLP solver through global resolution directly leads to efficiency gains in multi-step speculative decoding (which is what would be used in practice for real deployment).
>
> In addition, we have now implemented SpecTr: see our global comment (https://openreview.net/forum?id=gpsczXOsHn&noteId=a56AUrvD1D).
>
>
> Again, thank you for your thorough feedback and insightful comments. If you have any questions on any of our responses, or further suggestions on how to improve our work, please let us know. We are happy to continue discussion at any time until the end of the rebuttal period. We have made a significant effort to address each of your questions and weaknesses and would appreciate it if you would consider raising your score in light of our response. Thank you!

---

### Author Response · Authors · 2025-11-19
**Added Real-World Latency Experiments**

Thank you to all the reviewers for their considered feedback on our work. A majority of them have suggested that we implement global resolution in a realistic framework to measure end-to-end latency improvements in LLM decoding.

To this end, we have now implemented global resolution in SpecTr (https://arxiv.org/abs/2310.15141) to measure end-to-end latency in multi-step speculative decoding systems. SpecTr first drafts $K$ i.i.d. paths of length $L$ autoregressively from the draft model to form a draft tree, and then incrementally uses any OTLP solver (with $n$ being set to the number of child nodes) to advance from the root of the tree to child nodes, until it lands off the tree. We use global resolution for our OTLP solver (which is near-optimal) with $k=100$ for top-$k$ draft sampling, and when it fails, we resort to standard sampling from the target distribution $p$ (an approximate OTLP solver). We run our experiments on Gemma-27B/2B with temperature $1.0$ target sampling on the same dataset as our solve time experiments. Our results for $K=2,3,4$ and $L=8$ are shown below.


| K | Block efficiency (tokens/target call) | Walltime speedup (relative to vanilla) |
|-----|--------------------------------------|----------------------------------------|
| 2   | 2.23                                 | 1.84                                   |
| 3   | 2.54                                 | 1.89                                   |
| 4   | 2.65                                 | 1.98                                   |


Our walltime speedups in the best setting ($K=4$) are nearly $2\times$ better than vanilla decoding. **We also emphasize that these numbers are highly conservative estimates.** Unlike previous OTLP solvers used in the SpecTr framework, global resolution is situational: it is provably nearly optimal, but cannot always be used. This means the walltime and efficiency numbers above can be improved significantly by using other approximate OTLP solvers (like K-SEQ, the one used in the original SpecTr paper) when global resolution fails, rather than our simple $p$-sampling heuristic (which is a valid OTLP solver, but has poor acceptances). Furthermore, by modifying the choice of $k$ in top-$k$ draft sampling, one might be able to use global resolution to get high acceptances in failure scenarios for other $k$. We leave such selection of OTLP solvers and their parameters to future work.

Thank you again to all the reviewers for suggesting the above point, which we have now additionally incorporated into our draft in Appendix R.

---

### Author Response · Authors · 2025-12-02
**Meta-Rebuttal by Authors**

We thank the reviewers for their detailed suggestions on how to improve our work. We note that all reviewers highlight the soundness of our theoretical analysis on the optimal transport linear program (OTLP), and the empirical effectiveness of our global resolution solver. We have directly addressed all feedback from reviewers, including additional experiments and ablations, which we summarize below.

Reviewer **UvVU** asked about **empirical comparisons to other OTLP papers**, such as Hu et. al. (2025). We actually provided this in the original manuscript, and have since included additional clarification on where to find this (Appendix O).

Reviewer **UvVU** also raised important points about the **organization and presentation of the global resolution algorithm**, including a clarification about the **computational efficiency of the convex solver**. Accordingly, we updated our manuscript to include a succinct three-step decomposition of our algorithm at the beginning of Section 6: (1) solving for $H^\star$, (2) solving for outer residuals with our polymatroid approach, and (3) solving the convex systems. We explained how previous work only tackles (1) and does not result in a real verification algorithm, and we added discussion in Section 2 on why the single-step OTLP problem is important and how these solvers can be integrated into multi-step frameworks like SpecTr. On the computational efficiency point, we now explicitly clarify in Appendix K how we use PIE and group like terms to remove the $H^\star$ size dependence in the convex solver and make it run in milliseconds.

Reviewer **ZwDM** questioned the **robustness of global resolution to temperature variations**. In response, we ran both acceptance rate and global resolution ($\tau=10^{-4}$) solve time experiments for Gemma-27B/2B with temperatures 0.2, 0.4, 0.6, and 0.8 (all original results were for temperature 1.0), now included in Appendix Q. We found that at lower temperatures, increasing $k$ in top-$k$ sampling of the draft model offers little benefit to acceptance. This means our general solvers (LP, max-flow, optimized max-flow), which require $k$ to be small to remain tractable, can achieve high acceptance. The failure rate and solve times of global resolution also significantly decrease at low temperatures, naturally leading to better acceptance. At very low temperatures like 0.2, the performance gap between global resolution and general solvers diminishes. However, for moderate temperatures between 0.2 and 1.0, global resolution likely achieves the highest improvement over general solvers, as the overall failure rate remains low while still benefiting from higher acceptances with large $k$.

Reviewers **ZwDM** and **TpbB** also asked about **integration into real-world speculative decoding systems to measure end-to-end throughput**. To address this key point, as mentioned in our global comment, we have implemented global resolution in SpecTr, which can use any OTLP solver to form a valid multi-step speculative decoding algorithm. Our implementation achieves near $2\times$ decoding speedups, even when an inefficient $p$-sampling OTLP approximation is used in global resolution failure cases.

On the theoretical side of the multi-step SpecTr extension, following a question by reviewer **TpbB** on **compounding errors**, we have shown in Appendix K that the $L$-step approximation error scales at most linearly with $L$. In the evaluated $L=8$ setting with tolerance $\tau=10^{-4}$, the multi-step L1 error is concretely bounded above by $0.012$ at each iteration.

Finally, reviewers **YzHd** and **TpbB** both asked about the potential of **extending global resolution with approximation guarantees to other non-i.i.d. drafting settings**, such as independent but non-identical drafts and sampling without replacement. Following our three-step decomposition of global resolution in the response to reviewer **UvVU**, we added a discussion in Appendix S on which steps immediately or potentially generalize to other drafting methods. Sampling without replacement only requires modification to the approximation bounds, and $n=2$ independent non-identical drafts also appears tractable with some additional efficiency work on submodular minimization, but other settings are significantly more complicated and require new machinery.

We thank the reviewers again for their detailed suggestions on ablations, theory presentation and organization, and real-world evaluations. These have led to significant improvements in our experimental coverage and writing.

---

### Meta-Review · Area_Chair_8EMY · 2025-12-26

**Summary:**

The reviewers raised several concerns regarding the paper "Global Resolution: Optimal Multi-Draft Speculative Sampling via Convex Optimization."

Some reviewers noted that while the paper provides compelling results, there was a lack of empirical comparisons to other recent works, specifically mentioning Hu et al. (2025). They sought clarification on how the proposed algorithm stands against existing methods.

There were questions about the computational efficiency of the proposed convex minimization approach, particularly regarding the dependence on the vocabulary size and the steps involved in the algorithm. Reviewers requested more clarity on how the algorithm can maintain efficiency given these concerns.

Reviewers pointed out that there was insufficient exploration of the algorithm's robustness concerning temperature variations. They recommended conducting more experiments to assess performance under different temperature settings.

The reliance on i.i.d. drafts from a single model was noted as a limitation, with suggestions to explore generalizations to independent but non-identical drafts and sampling without replacement.

**Reviewer Concerns:**

The authors clarified that they provided comparisons with Hu et al. (2025) in the original manuscript and added further explanation in the rebuttal. They indicated that global resolution achieves higher acceptance rates compared to the greedy construction from Hu et al.

The rebuttal included a detailed explanation of how the proposed algorithm tackles computational efficiency concerns. The authors outlined their approach to reduce dependence on the vocabulary size, demonstrating that they can achieve efficient computation through specific strategies.

To address concerns about robustness under varying temperature settings, the authors ran additional experiments with different temperatures (0.2, 0.4, 0.6, and 0.8) and included these results in the revised manuscript, showing how acceptance rates and solve times varied accordingly.

**Reviewer Scores:**

The scores were already rather high. I believe the reviewers with scores 6 would have increased their scores to 8 as the authors did a pretty good job in responding to their concerns, especially wrt robustness under different temperature settings. The authors also made some clarifications about the computational efficiency of the convex solver and implemented global resolution in SpecTr, which can use any OTLP solver to form a valid multi-step speculative decoding algorithm.

---

### Decision · Program_Chairs · 2026-01-26

Accept (Oral)